# A Transformer-based Framework for Multivariate Time Series Representation Learning

## Abstract

In this work we propose for the first time a transformer-based framework for unsupervised representation learning of multivariate time series. Pre-trained models can be potentially used for downstream tasks such as regression and classification, forecasting and missing value imputation. We evaluate our models on several benchmark datasets for multivariate time series regression and classification and show that they exceed current state-of-the-art performance, even when the number of training samples is very limited, while at the same time offering computational efficiency. We show that unsupervised pre-training of our transformer models offers a substantial performance benefit over fully supervised learning, even without leveraging additional unlabeled data, i.e., by reusing the same data samples through the unsupervised objective.

## 1 Introduction

Multivariate time series (MTS) are an important type of data that is ubiquitous in a wide variety of domains, including science, medicine, finance, engineering and industrial applications. Despite the recent abundance of MTS data in the much touted era of "Big Data", the availability of *labeled* data in particular is far more limited: extensive data labeling is often prohibitively expensive or impractical, as it may require much time and effort, special infrastructure or domain expertise. For this reason, in all aforementioned domains there is great interest in methods which can offer high accuracy by using only a limited amount of labeled data or by leveraging the existing plethora of unlabeled data.

There is a large variety of modeling approaches for univariate and multivariate time series, with deep learning models recently challenging or replacing the state of the art in tasks such as forecasting, regression and classification (De Brouwer et al., 2019; Tan et al., 2020a; Fawaz et al., 2019b). However, unlike in domains such as Computer Vision or Natural Language Processing (NLP), the dominance of deep learning for time series is far from established: in fact, non-deep learning methods such as TS-CHIEF (Shifaz et al., 2020), HIVE-COTE (Lines et al., 2018), and ROCKET (Dempster et al., 2020) currently hold the record on time series regression and classification dataset benchmarks (Tan et al., 2020a; Bagnall et al., 2017), matching or even outperforming sophisticated deep architectures such as InceptionTime (Fawaz et al., 2019a) and ResNet (Fawaz et al., 2019b).

In this work, we investigate, for the first time, the use of a transformer encoder for unsupervised representation learning of multivariate time series, as well as for the tasks of time series regression and classification. Transformers are an important, recently developed class of deep learning models, which were first proposed for the task of natural language translation (Vaswani et al., 2017) but have since come to monopolize the state-of-the-art performance across virtually all NLP tasks (Raffel et al., 2019). A key factor for the widespread success of transformers in NLP is their aptitude for learning how to represent natural language through unsupervised pre-training (Brown et al., 2020; Raffel et al., 2019; Devlin et al., 2018). Besides NLP, transformers have also set the state of the art in several domains of sequence generation, such as polyphonic music composition (Huang et al., 2018).

Transformer models are based on a multi-headed attention mechanism that offers several key advantages and renders them particularly suitable for time series data (see Appendix section A.4 for details). Inspired by the impressive results attained through unsupervised pre-training of transformer models in NLP, as our main contribution, in the present work we develop a generally applicable

methodology (framework) that can leverage unlabeled data by first training a transformer model to extract dense vector representations of multivariate time series through an input denoising (autoregressive) objective. The pre-trained model can be subsequently applied to several downstream tasks, such as regression, classification, imputation, and forecasting. Here, we apply our framework for the tasks of multivariate time series regression and classification on several public datasets and demonstrate that transformer models can convincingly outperform all current state-of-the-art modeling approaches, even when only having access to a very limited amount of training data samples (on the order of hundreds of samples), an unprecedented success for deep learning models. Importantly, despite common preconceptions about transformers from the domain of NLP, where top performing models have billions of parameters and require days to weeks of pre-training on many parallel GPUs or TPUs, we also demonstrate that our models, using at most hundreds of thousands of parameters, can be trained even on CPUs, while training them on GPUs allows them to be trained as fast as even the fastest and most accurate non-deep learning based approaches.

## 2 RELATED WORK

**Regression and classification of time series**: Currently, non-deep learning methods such as TS-CHIEF (Shifaz et al., 2020), HIVE-COTE (Lines et al., 2018), and ROCKET (Dempster et al., 2020) constitute the state of the art for time series regression and classification based on evaluations on public benchmarks (Tan et al., 2020a; Bagnall et al., 2017), followed by CNN-based deep architectures such as InceptionTime (Fawaz et al., 2019a) and ResNet (Fawaz et al., 2019b). ROCKET, which on average is the best ranking method, is a fast method that involves training a linear classifier on top of features extracted by a flat collection of numerous and various random convolutional kernels. HIVE-COTE and TS-CHIEF (itself inspired by Proximity Forest (Lucas et al., 2019)), are very sophisticated methods which incorporate expert insights on time series data and consist of large, heterogeneous ensembles of classifiers utilizing shapelet transformations, elastic similarity measures, spectral features, random interval and dictionary-based techniques; however, these methods are highly complex, involve significant computational cost, cannot benefit from GPU hardware and scale poorly to datasets with many samples and long time series; moreover, they have been developed for and only been evaluated on *univariate* time series.

**Unsupervised learning for multivariate time series**: Recent work on unsupervised learning for multivariate time series has predominantly employed autoencoders, trained with an input reconstruction objective and implemented either as Multi-Layer Perceptrons (MLP) or RNN (most commonly, LSTM) networks. As interesting variations of the former, Kopf et al. (2019) and Fortuin et al. (2019) additionally incorporated Variational Autoencoding into this approach, but focused on clustering and the visualization of shifting sample topology with time. As an example of the latter, Malhotra et al. (2017) presented a multi-layered RNN sequence-to-sequence autoencoder, while Lyu et al. (2018) developed a multi-layered LSTM with an attention mechanism and evaluated both an input reconstruction (autoencoding) as well as a forecasting loss for unsupervised representation learning of Electronic Healthcare Record multivariate time series.

As a novel take on autoencoding, and with the goal of dealing with missing data, Bianchi et al. (2019) employ a stacked bidirectional RNN encoder and stacked RNN decoder to reconstruct the input, and at the same time use a user-provided kernel matrix as prior information to condition internal representations and encourage learning similarity-preserving representations of the input. They evaluate the method on the tasks of missing value imputation and classification of time series under increasing "missingness" of values.

A distinct approach is followed by Zhang et al. (2019), who use a composite convolutional - LSTM network with attention and a loss which aims at reconstructing correlation matrices between the variables of the multivariate time series input. They use and evaluate their method only for the task of anomaly detection.

Finally, Jansen et al. (2018) rely on a triplet loss and the idea of temporal proximity (the loss rewards similarity of representations between proximal segments and penalizes similarity between distal segments of the time series) for unsupervised representation learning of non-speech audio data. This idea is explored further by Franceschi et al. (2019), who combine the triplet loss with a deep causal dilated CNN, in order to make the method effective for very long time series.

**Transformer models for time series**: Recently, a full encoder-decoder transformer architecture was employed for *univariate* time series forecasting: Li et al. (2019) showed superior performance compared to the classical statistical method ARIMA, the recent matrix factorization method TRMF, an RNN-based autoregressive model (DeepAR) and an RNN-based state space model (DeepState) on 4 public forecasting datasets, while Wu et al. (2020) used a transformer to forecast influenza prevalence and similarly showed performance benefits compared to ARIMA, an LSTM and a GRU Seq2Seq model with attention, and Lim et al. (2020) used a transformer for multi-horizon univariate forecasting, supporting interpretation of temporal dynamics. Finally, Ma et al. (2019) use an encoder-decoder architecture with a variant of self-attention for imputation of missing values in multivariate, geo-tagged time series and outperform classic as well as the state-of-the-art, RNN-based imputation methods on 3 public and 2 competition datasets for imputation.

By contrast, our work aspires to generalize the use of transformers from solutions to specific generative tasks (which require the full encoder-decoder architecture) to a framework which allows for unsupervised pre-training and with minor modifications can be readily used for a wide variety of downstream tasks; this is analogous to the way BERT (Devlin et al., 2018) converted a translation model into a generic framework based on unsupervised learning, an approach which has become a de facto standard and established the dominance of transformers in NLP.

## 3 METHODOLOGY

### 3.1 BASE MODEL

At the core of our method lies a transformer encoder, as described in the original transformer work by Vaswani et al. (2017); however, we do not use the decoder part of the architecture. A schematic diagram of the generic part of our model, common across all considered tasks, is shown in Figure 1. We refer the reader to the original work for a detailed description of the transformer model, and here present the proposed changes that make it compatible with multivariate time series data, instead of sequences of discrete word indices.

In particular, each training sample $\mathbf{X} \in \mathbb{R}^{w \times m}$, which is a multivariate time series of length $w$ and $m$ different variables, constitutes a sequence of $w$ feature vectors $\mathbf{x_t} \in \mathbb{R}^m$: $\mathbf{X} \in \mathbb{R}^{w \times m} = [\mathbf{x_1}, \mathbf{x_2}, \ldots, \mathbf{x_w}]$. The original feature vectors $\mathbf{x_t}$ are first normalized (for each dimension, we subtract the mean and divide by the variance across the training set samples) and then linearly projected onto a $d$-dimensional vector space, where $d$ is the dimension of the transformer model sequence element representations (typically called *model dimension*):

$$\mathbf{u_t} = \mathbf{W_p} \mathbf{x_t} + \mathbf{b_p} \tag{1}$$

where $\mathbf{W_p} \in \mathbb{R}^{d \times m}$, $\mathbf{b_p} \in \mathbb{R}^d$ are learnable parameters and $\mathbf{u_t} \in \mathbb{R}^d, t = 0, \ldots, w$ are the model input vectors[1]. These will become the queries, keys and values of the self-attention layer, after adding the positional encodings and multiplying by the corresponding matrices.

We note that the above formulation also covers the univariate time series case, i.e., $m = 1$, although we only evaluate our approach on multivariate time series in the scope of this work. We additionally note that the input vectors $\mathbf{u_t}$ need not necessarily be obtained from the (transformed) feature vectors at a time step $t$: because the computational complexity of the model scales as $O(w^2)$ and the number of parameters[2] as $O(w)$ with the input sequence length $w$, to obtain $\mathbf{u_t}$ in case the granularity (temporal resolution) of the data is very fine, one may instead use a 1D-convolutional layer with 1 input and $d$ output channels and kernels $K_i$ of size $(k, m)$, where $k$ is the width in number of time steps and $i$ the output channel:

$$u_t{}^i = u(t, i) = \sum_j \sum_h x(t + j, h) K_i(j, h), \quad i = 1, \ldots, d \tag{2}$$

---

[1] Although equation 1 shows the operation for a single time step for clarity, all input vectors are embedded concurrently by a single matrix-matrix multiplication

[2] Specifically, the learnable positional encoding, batch normalization and output layers

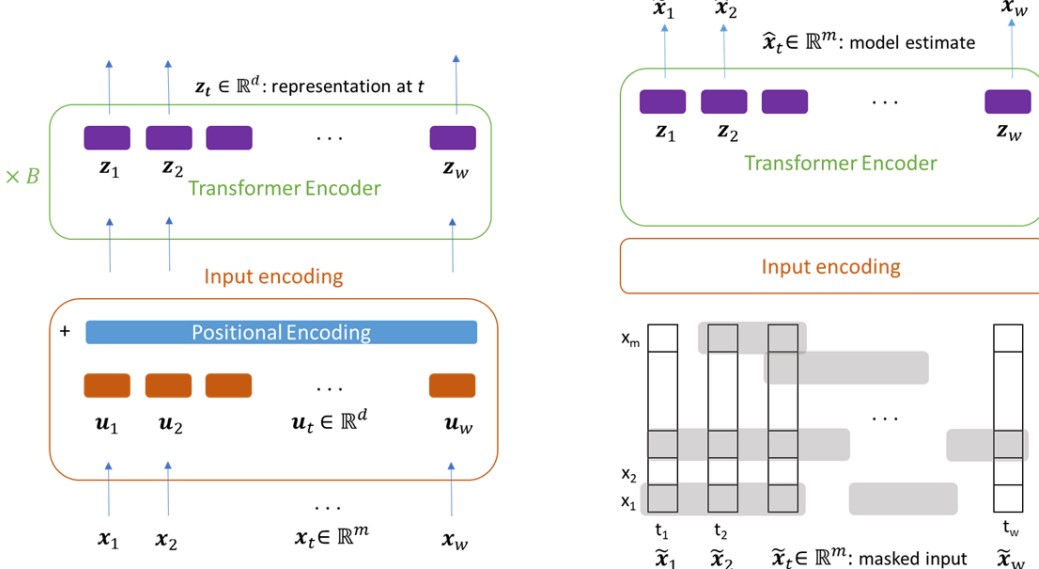

Figure 1: **Left:** Generic model architecture, common to all tasks. The feature vector $\mathbf{x_t}$ at each time step $t$ is linearly projected to a vector $\mathbf{u_t}$ of the same dimensionality $d$ as the internal representation vectors of the model and is fed to the first self-attention layer to form the keys, queries and values after adding a positional encoding. **Right:** Training setup of the unsupervised pre-training task. We mask a proportion $r$ of each variable sequence in the input independently, such that across each variable, time segments of mean length $l_m$ are masked, each followed by an unmasked segment of mean length $l_u = \frac{1-r}{r} l_m$. Using a linear layer on top of the final vector representations $\mathbf{z_t}$, at each time step the model tries to predict the full, uncorrupted input vectors $\mathbf{x_t}$; however, only the predictions on the masked values are considered in the Mean Squared Error loss.

In this way, one may control the temporal resolution by using a stride or dilation factor greater than 1. Moreover, although in the present work we only used equation 1, one may use equation 2 as an input to compute the keys and queries and equation 1 to compute the values of the self-attention layer. This is particularly useful in the case of univariate time series, where self-attention would otherwise match (consider relevant/compatible) all time steps which share similar values for the independent variable, as noted by Li et al. (2019).

Finally, since the transformer is a feed-forward architecture that is insensitive to the ordering of input, in order to make it aware of the sequential nature of the time series, we add positional encodings $W_{\text{pos}} \in \mathbb{R}^{w \times d}$ to the input vectors $U \in \mathbb{R}^{w \times d} = [\mathbf{u_1}, \dots, \mathbf{u_w}]$: $U' = U + W_{\text{pos}}$.

Instead of deterministic, sinusoidal encodings, which were originally proposed by Vaswani et al. (2017), we use fully learnable positional encodings, as we observed that they perform better for all datasets presented in this work. Based on the performance of our models, we also observe that the positional encodings generally appear not to significantly interfere with the numerical information of the time series, similar to the case of word embeddings; we hypothesize that this is because they are learned so as to occupy a different, approximately orthogonal, subspace to the one in which the projected time series samples reside. This approximate orthogonality condition is much easier to satisfy in high dimensional spaces.

An important consideration regarding time series data is that individual samples may display considerable variation in length. This issue is effectively dealt with in our framework: after setting a maximum sequence length $w$ for the entire dataset, shorter samples are padded with arbitrary values, and we generate a padding mask which adds a large negative value to the attention scores for the padded positions, before computing the self-attention distribution with the softmax function. This forces the model to completely ignore padded positions, while allowing the parallel processing of samples in large minibatches.

Transformers in NLP use layer normalization after computing self-attention and after the feed-forward part of each encoder block, leading to significant performance gains over batch normalization, as originally proposed by Vaswani et al. (2017). However, here we instead use batch normalization, because it can mitigate the effect of outlier values in time series, an issue that does not arise in NLP word embeddings. Additionally, the inferior performance of batch normalization in NLP has been mainly attributed to extreme variation in sample length (i.e., sentences in most tasks) (Shen et al., 2020), while in the datasets we examine this variation is much smaller. In Table 11 of the Appendix we show that batch normalization can indeed offer a very significant performance benefit over layer normalization, while the extent can vary depending on dataset characteristics.

## 3.2 REGRESSION AND CLASSIFICATION

The base model architecture presented in Section 3.1 and depicted in Figure 1 can be used for the purposes of regression and classification with the following modification: the final representation vectors $\mathbf{z_t} \in \mathbb{R}^d$ corresponding to all time steps are concatenated into a single vector $\bar{\mathbf{z}} \in \mathbb{R}^{d \cdot w} = [\mathbf{z_1}; \ldots; \mathbf{z_w}]$, which serves as the input to a linear output layer with parameters $\mathbf{W_o} \in \mathbb{R}^{n \times (d \cdot w)}$, $\mathbf{b_o} \in \mathbb{R}^n$, where $n$ is the number of scalars to be estimated for the regression problem (typically $n = 1$), or the number of classes for the classification problem:

$$\hat{\mathbf{y}} = \mathbf{W_o}\bar{\mathbf{z}} + \mathbf{b_o} \tag{3}$$

In the case of regression, the loss for a single data sample will simply be the squared error $\mathcal{L} = \|\hat{\mathbf{y}} - \mathbf{y}\|^2$, where $\mathbf{y} \in \mathbb{R}^n$ are the ground truth values. We clarify that regression in the context of this work means predicting a numeric value for a given sequence (time series sample). This numeric value is of a different nature than the numerical data appearing in the time series: for example, given a sequence of simultaneous temperature and humidity measurements of 9 rooms in a house, as well as weather and climate data such as temperature, pressure, humidity, wind speed, visibility and dewpoint, we wish to predict the total energy consumption in kWh of a house for that day. The parameter $n$ corresponds to the number of scalars (or the dimensionality of a vector) to be estimated.

In the case of classification, the predictions $\hat{\mathbf{y}}$ will additionally be passed through a softmax function to obtain a distribution over classes, and its cross-entropy with the categorical ground truth labels will be the sample loss.

Finally, when fine-tuning the pre-trained models, we allow training of all weights; instead, freezing all layers except for the output layer would be equivalent to using static, pre-extracted time-series representations of the time series. In Table 12 in the Appendix we show the trade-off in terms of speed and performance when using a fully trainable model versus static representations.

## 3.3 UNSUPERVISED PRE-TRAINING

As a task for the unsupervised pre-training of our model we consider the autoregressive task of denoising the input: specifically, we set part of the input to 0 and ask the model to predict the masked values. The corresponding setup is depicted in the right part of Figure 1. A binary noise mask $\mathbf{M} \in \mathbb{R}^{w \times m}$, is created independently for each training sample, and the input is masked by elementwise multiplication: $\tilde{\mathbf{X}} = \mathbf{M} \odot \mathbf{X}$. On average, a proportion $r$ of each mask column of length $w$ (corresponding to a single variable in the multivariate time series) is set to 0 by alternating between segments of 0s and 1s. We choose the state transition probabilities such that each masked segment (sequence of 0s) has a length that follows a geometric distribution with mean $l_m$ and is succeeded by an unmasked segment (sequence of 1s) of mean length $l_u = \frac{1-r}{r}l_m$. We chose $l_m = 3$ for all presented experiments. The reason why we wish to control the length of the masked sequence, instead of simply using a Bernoulli distribution with parameter $r$ to set all mask elements independently at random, is that very short masked sequences (e.g., of 1 masked element) in the input can often be trivially predicted with good approximation by replicating the immediately preceding or succeeding values or by the average thereof. In order to obtain enough long masked sequences with relatively high likelihood, a very high masking proportion $r$ would be required, which would render the overall task detrimentally challenging. Following the process above, at each time step on average $r \cdot m$ variables will be masked. We empirically chose $r = 0.15$ for all presented experiments. This input masking process is different from the "cloze type" masking used by NLP models such as BERT,

where a special token and thus word embedding vector replaces the original word embedding, i.e., the *entire* feature vector at affected time steps. We chose this masking pattern because it encourages the model to learn to attend both to preceding and succeeding segments in individual variables, as well as to existing contemporary values of the other variables in the time series, and thereby to learn to model inter-dependencies between variables. In Table 10 in the Appendix we show that this masking scheme is more effective than other possibilities for denoising the input.

Using a linear layer with parameters $\mathbf{W_o} \in \mathbb{R}^{m \times d}$, $\mathbf{b_o} \in \mathbb{R}^m$ on top of the final vector representations $\mathbf{z_t} \in \mathbb{R}^d$, for each time step the model concurrently outputs its estimate $\hat{\mathbf{x}}_\mathbf{t}$ of the full, uncorrupted input vectors $\mathbf{x_t}$; however, only the predictions on the masked values (with indices in the set $M \equiv \{(t,i) : m_{t,i} = 0\}$, where $m_{t,i}$ are the elements of the mask $\mathbf{M}$), are considered in the Mean Squared Error loss for each data sample:

$$\hat{\mathbf{x}}_\mathbf{t} = \mathbf{W_o z_t} + \mathbf{b_o} \tag{4}$$

$$\mathcal{L}_{\text{MSE}} = \frac{1}{|M|} \sum_{(t,i) \in M} \sum \left( \hat{x}(t,i) - x(t,i) \right)^2 \tag{5}$$

This objective differs from the one used by denoising autoencoders, where the loss considers reconstruction of the entire input, under (typically Gaussian) noise corruption. Also, we note that the approach described above differs from simple dropout on the input embeddings, both with respect to the statistical distributions of masked values, as well as the fact that here the masks also determine the loss function. In fact, we additionally use a dropout of 10% when training all of our supervised and unsupervised models.

## 4 EXPERIMENTS & RESULTS

In the experiments reported below we use the predefined training - test set splits of the benchmark datasets and train all models long enough to ensure convergence. We do this to account for the fact that training the transformer models in a fully supervised way typically requires more epochs than fine-tuning the ones which have already been pre-trained using the unsupervised methodology of Section 3.3. Because the benchmark datasets are very heterogeneous in terms of number of samples, dimensionality and length of the time series, as well as the nature of the data itself, we observed that we can obtain better performance by a cursory tuning of hyperparameters (such as the number of encoder blocks, the representation dimension, number of attention heads or dimension of the feedforward part of the encoder blocks) separately for each dataset. To select hyperparameters, for each dataset we randomly split the training set in two parts, 80%-20%, and used the 20% as a validation set for hyperparameter tuning. After fixing the hyperparameters, the entire training set was used to train the model again, which was finally evaluated on the test set. A set of hyperparameters which has consistently good performance on all datasets is shown in Table 14 in the Appendix, alongside the hyperparameters that we have found to yield the best performance for each dataset (Tables 15, 16, 17, 18.

### 4.1 REGRESSION

We select a diverse range of 6 datasets from the Monash University, UEA, UCR Time Series Regression Archive Tan et al. (2020a) in a way so as to ensure diversity with respect to the dimensionality and length of time series samples, as well as the number of samples (see Appendix Table 3 for dataset characteristics). Table 1 shows the Root Mean Squared Error achieved by of our models, named TST for "Time Series Transformer", including a variant trained only through supervision, and one first pre-trained on the same *training set* in an unsupervised way. We compare them with the currently best performing models as reported in the archive. Our transformer models rank first on all but two of the examined datasets, for which they rank second. They thus achieve an average rank of 1.33, setting them clearly apart from all other models; the overall second best model, XG-Boost, has an average rank of 3.5, ROCKET (which outperformed ours on one dataset) on average ranks in 5.67th place and Inception (which outperformed ours on the second dataset) also has an average rank of 5.67. On average, our models attain 30% lower RMSE than the mean RMSE among

all models, and approx. 16% lower RMSE than the overall second best model (XGBoost), with absolute improvements varying among datasets from approx. 4% to 36%. We note that all other deep learning methods achieve performance close to the middle of the ranking or lower. In Table 1 we report the "average relative difference from mean" metric $r_j$ for each model $j$, the over $N$ datasets:

$$r_j = \frac{1}{N} \sum_{i=1}^{N} \frac{R(i,j) - \bar{R}_i}{\bar{R}_i}, \quad \bar{R}_i = \frac{1}{M} \sum_{k=1}^{M} R(i,k)$$

, where $R(i,j)$ is the RMSE of model $j$ on dataset $i$ and $M$ is the number of models.

Importantly, we also observe that the pre-trained transformer models outperform the fully supervised ones in 3 out of 6 datasets. This is interesting, because no additional samples are used for pre-training: the benefit appears to originate from reusing the same training samples for learning through an unsupervised objective. To further elucidate this observation, we investigate the following questions:

**Q1: Given a partially labeled dataset of a certain size, how will additional labels affect performance?** This pertains to one of the most important decisions that data owners face, namely, to what extent will further annotation help. To clearly demonstrate this effect, we choose the largest dataset we have considered from the regression archive (12.5k samples), in order to avoid the variance introduced by small set sizes. The left panel of Figure 2 (where each marker is an experiment) shows how performance on the entire test set varies with an increasing proportion of labeled training set data used for *supervised* learning. As expected, with an increasing proportion of available labels performance improves both for a fully supervised model, as well as the same model that has been first pre-trained on the entire training set through the unsupervised objective and then fine-tuned. Interestingly, not only does the pre-trained model outperform the fully supervised one, but the benefit persists throughout the entire range of label availability, even when the models are allowed to use all labels; this is consistent with our previous observation on Table 1 regarding the advantage of reusing samples.

**Q2: Given a labeled dataset, how will additional *unlabeled* samples affect performance?** In other words, to what extent does unsupervised learning make it worth collecting more data, even if no additional annotations are available? This question differs from the above, as we now only scale the availability of data samples for *unsupervised* pre-training, while the number of labeled samples is fixed. The right panel of Figure 2 (where each marker is an experiment) shows that, for a given number of labels (shown as a percentage of the totally available labels), the more data samples are used for unsupervised learning, the lower the error achieved (note that the horizontal axis value 0 corresponds to fully supervised training only, while all other values to unsupervised pre-training followed by supervised fine-tuning). This trend is more linear in the case of supervised learning on 20% of the labels (approx. 2500). Likely due to a small sample (here, meaning set) effect, in the case of having only 10% of the labels (approx. 1250) for supervised learning, the error first decreases rapidly as we use more samples for unsupervised pre-training, and then momentarily increases, before it decreases again (for clarity, the same graphs are shown separately in Figure 3 in the Appendix). Consistent with our observations above, it is interesting to again note that, for a given number of labeled samples, even reusing a subset of the *same samples* for unsupervised pre-training improves performance: for the 1250 labels (blue diamonds of the right panel of Figure 2 or left panel of Figure 3 in the Appendix) this can be observed in the horizontal axis range $[0, 0.1]$, and for the 2500 labels (blue diamonds of the right panel of Figure 2 or right panel of Figure 3 in the Appendix) in the horizontal axis range $[0, 0.2]$.

### 4.2 CLASSIFICATION

We select a set of 11 multivariate datasets from the UEA Time Series Classification Archive (Bagnall et al., 2018) with diverse characteristics in terms of the number, dimensionality and length of time series samples, as well as the number of classes (see Appendix Table 4). As this archive is new, there have not been many reported model evaluations; we follow Franceschi et al. (2019) and use as a baseline the best performing method studied by the creators of the archive, $DTW_D$ (dimension-Dependent DTW), together with the method proposed by Franceschi et al. (2019) themselves (a dilation-CNN leveraging unsupervised and supervised learning). Additionally, we use the publicly available implementations Tan et al. (2020b) of ROCKET, which is currently the top performing

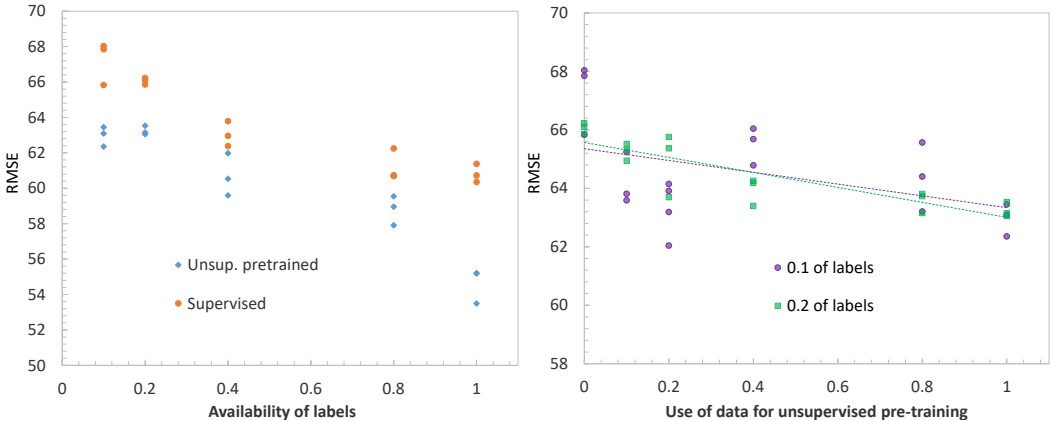

Figure 2: Dataset: BeijingPM25Quality. **Left:** Root Mean Squared Error of a fully supervised transformer (orange circles) and the same model pre-trained (blue diamonds) on the training set through the unsupervised objective and then fine-tuned on available labels, versus the proportion of labeled data in the training set. **Right:** Root Mean Squared Error of a given model as a function of the number of samples (here, shown as a proportion of the total number of samples in the training set) used for unsupervised pre-training. For supervised learning, two levels of label availability are depicted: 10% (purple circles) and 20% (green squares) of all training data labels. Note that a horizontal axis value of 0 means fully supervised learning only, while all other values correspond to unsupervised pre-training followed by supervised fine-tuning.

| | | | | | | | | | | | | Ours | |
|---|---|---|---|---|---|---|---|---|---|---|---|---|---|
| | | | | | | | Root MSE | | | | | | |
| **Dataset** | **SVR** | **Random Forest** | **XGBoost** | **1-NN-ED** | **5-NN -ED** | **1-NN-DTWD** | **5-NN-DTWD** | **Rocket** | **FCN** | **ResNet** | **Inception** | **TST (sup. only)** | **TST (pretrained)** |
| AppliancesEnergy | 3.457 | 3.455 | 3.489 | 5.231 | 4.227 | 6.036 | 4.019 | 2.299 | 2.865 | 3.065 | 4.435 | **2.228** | 2.375 |
| BenzeneConcentr. | 4.790 | 0.855 | 0.637 | 6.535 | 5.844 | 4.983 | 4.868 | 3.360 | 4.988 | 4.061 | 1.584 | 0.517 | **0.494** |
| BeijingPM10 | 110.574 | 94.072 | 93.138 | 139.229 | 115.669 | 139.134 | 115.502 | 120.057 | 94.348 | 95.489 | 96.749 | 91.344 | **86.866** |
| BeijingPM25 | 75.734 | 63.301 | 59.495 | 88.193 | 74.156 | 88.256 | 72.717 | 62.769 | 59.726 | 64.462 | 62.227 | 60.357 | **53.492** |
| LiveFuelMoisture | 43.021 | 44.657 | 44.295 | 58.238 | 46.331 | 57.111 | 46.290 | **41.829** | 47.877 | 51.632 | 51.539 | 42.607 | 43.138 |
| IEEEPPG | 36.301 | 32.109 | 31.487 | 33.208 | 27.111 | 37.140 | 33.572 | 36.515 | 34.325 | 33.150 | **23.903** | 25.042 | 27.806 |
| **Avg Rel. diff. from mean** | 0.097 | -0.172 | -0.197 | 0.377 | 0.152 | 0.353 | 0.124 | -0.048 | 0.021 | 0.005 | -0.108 | -0.301 | **-0.303** |
| **Avg Rank** | 7.166 | 4.5 | 3.5 | 10.833 | 8 | 11.167 | 7.667 | 5.667 | 6.167 | 6.333 | 5.666 | **1.333** | |

Table 1: Performance on **multivariate regression** datasets, in terms of Root Mean Squared Error. Bold indicates best values, underlining indicates second best.

model for univariate time series and one of the best in our regression evaluation, and XGBoost, which is one of the most commonly used models for univariate and multivariate time series, and also the best baseline model in our regression evaluation (Section 4.1). Finally, we did not find any reported evaluations of RNN-based models on any of the UCR/UEA archives, possibly because of a common perception for long training and inference times, as well as difficulty in training (Fawaz et al., 2019b); therefore, we implemented a stacked LSTM model and also include it in the comparison. The performance of the baselines alongside our own models are shown in Table 2 in terms of accuracy, to allow comparison with reported values.

It can be seen that our models performed best on 7 out of the 11 datasets, achieving an average rank of 1.7, followed by ROCKET, which performed best on 3 datasets and on average ranked 2.3th. The dilation-CNN (Franceschi et al., 2019) and XGBoost, which performed best on the remaining 1 dataset, tied and on average ranked 3.7th and 3.8th respectively. Interestingly, we observe that all datasets on which ROCKET outperformed our model were very low dimensional (specifically, 3-dimensional). Although our models still achieved the second best performance for UWaveG-estureLibrary, in general we believe that this indicates a relative weakness of our current models when dealing with very low dimensional time series. As discussed in Section 3.1, this may be due to the problems introduced by a low-dimensional representation space to the attention mechanism, as well as the added positional embeddings; to mitigate this issue, in future work we intend to use a 1D-convolutional layer to extract more meaningful representations of low-dimensional input features (see Section 3.1). Conversely, our models performed particularly well on very high-

| Dataset | Ours | | Rocket | XGBoost | LSTM | Frans. et al | DTW_D |
|---|---|---|---|---|---|---|---|
| | TST (pretrained) | TST (sup. only) | | | | | |
| EthanolConcentration | 0.326 | 0.337 | **0.452** | 0.437 | 0.323 | 0.289 | 0.323 |
| FaceDetection | **0.689** | 0.681 | 0.647 | 0.633 | 0.577 | 0.528 | 0.529 |
| Handwriting | 0.359 | 0.305 | **0.588** | 0.158 | 0.152 | 0.533 | 0.286 |
| Heartbeat | **0.776** | **0.776** | 0.756 | 0.732 | 0.722 | 0.756 | 0.717 |
| JapaneseVowels | **0.997** | 0.994 | 0.962 | 0.865 | 0.797 | 0.989 | 0.949 |
| InsectWingBeat | **0.687** | 0.684 | - | 0.369 | 0.176 | 0.16 | - |
| PEMS-SF | 0.896 | 0.919 | 0.751 | **0.983** | 0.399 | 0.688 | 0.711 |
| SelfRegulationSCP1 | 0.922 | **0.925** | 0.908 | 0.846 | 0.689 | 0.846 | 0.775 |
| SelfRegulationSCP2 | **0.604** | 0.589 | 0.533 | 0.489 | 0.466 | 0.556 | 0.539 |
| SpokenArabicDigits | **0.998** | 0.993 | 0.712 | 0.696 | 0.319 | 0.956 | 0.963 |
| UWaveGestureLibrary | 0.913 | 0.903 | **0.944** | 0.759 | 0.412 | 0.884 | 0.903 |
| **Avg Accuracy** (excl. InsectWingBeat) | **0.748** | 0.742 | 0.725 | 0.659 | 0.486 | 0.703 | 0.669 |
| **Avg Rank** | 1.7 | | 2.3 | 3.8 | 5.4 | 3.7 | 4.1 |

Table 2: Accuracy on **multivariate classification** datasets. Bold indicates best and underlining second best values. A dash indicates that the corresponding method failed to run on this dataset.

dimensional datasets (FaceDetection, HeartBeat, InsectWingBeat, PEMS-SF), and/or datasets with relatively more training samples. As a characteristic example, on InsectWingBeat (which is by far the largest dataset with 30k samples and contains time series of 200 dimensions and highly irregular length) our model reached an accuracy of 0.689, while all other methods performed very poorly - the second best was XGBoost with an accuracy of 0.369. However, we note that our model performed exceptionally well also on datasets with only a couple of hundred samples, which in fact constitute 8 out of the 11 examined datasets.

Finally, we observe that the pre-trained transformer models performed better than the fully supervised ones in 8 out of 11 datasets, sometimes by a substantial margin.Again, no additional samples were available for unsupervised pre-training, so the benefit appears to originate from reusing the same samples.

## 5 CONCLUSION

In this work we propose a novel framework for multivariate time series representation learning based on the transformer encoder architecture. The framework includes an unsupervised pre-training scheme, which we show that can offer substantial performance benefits over fully supervised learning, even without leveraging additional unlabeled data, i.e., by reusing the same data samples. By evaluating our framework on several public multivariate time series datasets from various domains and with diverse characteristics, we demonstrate that it is currently the best performing method for regression and classification, even for datasets where only a few hundred training samples are available.

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

# A  APPENDIX

## A.1  ADDITIONAL POINTS & FUTURE WORK

**Execution time for training:** While a precise comparison in terms of training time is well out of scope for the present work, in Section A.3 of the Appendix we demonstrate that our transformer-based method is economical in terms of its use of computational resources. However, alternative self-attention schemes, such as sparse attention patterns (Li et al., 2019), recurrence (Dai et al., 2019) or compressed (global-local) attention (Beltagy et al., 2020), can help drastically reduce the $O(w^2)$ complexity of the self-attention layers with respect to the time series length $w$, which is the main performance bottleneck.

**Imputation and forecasting:** The model and training process described in Section 3.3 is exactly the setup required to perform imputation of missing values, without any modifications, and we observed that it was possible to achieve very good results following this method; as a rough indication, our models could reach Root Mean Square Errors very close to 0 when asked to perform the input denoising (autoregressive) task on the test set, after being subjected to unsupervised pre-training on the training set. We also show example results of imputation on one of the datasets presented in this work in Figure 5. However, we defer a systematic quantitative comparison with the state of the art to future work. Furthermore, we note that one may simply use different patterns of masking to achieve different objectives, while the rest of the model and setup remain the same. For example, using a mask which conceals the last part of all variables simultaneously, one may perform forecasting (see Figure 4 in Appendix), while for longer time series one may additionally perform this process within a sliding window. Again, we defer a systematic investigation to future work.

**Extracted representations:** The representations $\mathbf{z_t}$ extracted by the transformer models can be used directly for evaluating similarity between time series, clustering, visualization and any other use cases where time series representations are used in practice. A valuable benefit offered by transformers is that representations can be independently addressed for each time step; this means that, for example, a greater weight can be placed at the beginning, middle or end of the time series, which allows to selectively compare time series, visualize temporal evolution of samples etc.

| Dataset | Train Size | Test Size | Length | Dimension | Missing Values |
|---|---|---|---|---|---|
| AppliancesEnergy | 96 | 42 | 144 | 24 | No |
| BenzeneConcentration | 3433 | 5445 | 240 | 8 | Yes |
| BeijingPM10Quality | 12432 | 5100 | 24 | 9 | Yes |
| BeijingPM25Quality | 12432 | 5100 | 24 | 9 | Yes |
| LiveFuelMoistureContent | 3493 | 1510 | 365 | 7 | No |
| IEEEPPG | 1768 | 1328 | 1000 | 5 | No |

Table 3: Multivariate Regression Datasets

| Dataset | TrainSize | TestSize | NumDimensions | SeriesLength | NumClasses |
|---|---|---|---|---|---|
| EthanolConcentration | 261 | 263 | 3 | 1751 | 4 |
| FaceDetection | 5890 | 3524 | 144 | 62 | 2 |
| Handwriting | 150 | 850 | 3 | 152 | 26 |
| Heartbeat | 204 | 205 | 61 | 405 | 2 |
| InsectWingbeat | 30000 | 20000 | 200 | 30 | 10 |
| JapaneseVowels | 270 | 370 | 12 | 29 | 9 |
| PEMS-SF | 267 | 173 | 963 | 144 | 7 |
| SelfRegulationSCP1 | 268 | 293 | 6 | 896 | 2 |
| SelfRegulationSCP2 | 200 | 180 | 7 | 1152 | 2 |
| SpokenArabicDigits | 6599 | 2199 | 13 | 93 | 10 |
| UWaveGestureLibrary | 120 | 320 | 3 | 315 | 8 |

Table 4: Multivariate Classification Datasets

| Dataset | Standard deviation | |
|---|---|---|
| | Supervised TST | Pre-trained TST |
| AppliancesEnergy | 0.240 | 0.163 |
| BenzeneConcentration | 0.031 | 0.092 |
| BeijingPM10Quality | 0.689 | 0.813 |
| BeijingPM25Quality | 0.189 | 0.253 |
| LiveFuelMoistureContent | 0.735 | 0.013 |
| IEEEPPG | 1.079 | 1.607 |

Table 5: Standard deviation of the Root Mean Square Error displayed by the Time Series Transformer models on multivariate regression datasets

### A.2 CRITERIA FOR DATASET SELECTION

We select a diverge range of datasets from the Monash University, UEA, UCR Time Series Regression and Classification Archives, in a way so as to ensure diversity with respect to the dimensionality and length of time series samples, as well as the number of samples. Additionally, we have tried to include both "easy" and "difficult" datasets (where the baselines perform very well or less well). In the following we provide a more detailed rationale for each of the selected multivariate datasets.

**EthanolConcentration**: very low dimensional, very few samples, moderate length, large number of classes, challenging

**FaceDetection**: very high dimensional, many samples, very short length, minimum number of classes

**Handwriting**: very low dimensional, very few samples, moderate length, large number of classes

**Heartbeat**: high dimensional, very few samples, moderate length, minimum number of classes

**JapaneseVowels**: very heterogeneous sample length, moderate num. dimensions, very few samples, very short length, moderate number of classes, all baselines perform well

**InsectWingBeat**: very high dimensional, many samples, very short length, moderate number of classes, very challenging

**PEMS-SF**: extremely high dimensional, very few samples, moderate length, moderate number of classes

**SelfRegulationSCP1**: Few dimensions, very few samples, long length, minimum number of classes, baselines perform well

**SelfRegulationSCP2**: similar to SelfRegulationSCP1, but challenging

**SpokenArabicDigits**: Moderate number of dimensions, many samples, very heterogeneous length, moderate number of classes, most baselines perform well

**UWaveGestureLibrary**: very low dimensional, very few samples, moderate length, moderate number of classes, baselines perform well

### A.3 EXECUTION TIME

We recorded the times required for training our fully supervised models until convergence on a GPU, as well as for the currently fastest and top performing (in terms of classification accuracy and regression error) baseline methods, ROCKET and XGBoost on a CPU. These have been shown to be orders of magnitude faster than methods such as TS-CHIEF, Proximity Forest, Elastic Ensembles, DTW and HIVE-COTE, but also deep learning based methods Dempster et al. (2020). Although XGBoost and ROCKET are incomparably faster than the transformer on a CPU, as can be seen in Table 7 in the Appendix, exploiting commercial GPUs and the parallel processing capabilities of a transformer typically enables as fast (and sometimes faster) training times as these (currently fastest available) methods. In practice, despite allowing for many hundreds of epochs, using a GPU we never trained our models longer than 3 hours on any of the examined datasets.

| Dataset | Standard deviation | |
| --- | --- | --- |
| | Supervised TST | Pre-trained TST |
| EthanolConcentration | 0.024 | 0.002 |
| FaceDetection | 0.007 | 0.006 |
| Handwriting | 0.020 | 0.006 |
| Heartbeat | 0.018 | 0.018 |
| InsectWingbeat | 0.003 | 0.026 |
| JapaneseVowels | 0.000 | 0.0016 |
| PEMS-SF | 0.017 | 0.003 |
| SelfRegulationSCP1 | 0.005 | 0.006 |
| SelfRegulationSCP2 | 0.020 | 0.003 |
| SpokenArabicDigits | 0.0003 | 0.001 |
| UWaveGestureLibrary | 0.005 | 0.003 |

Table 6: Standard deviation of accuracy displayed by the Time Series Transformer models on multivariate classification datasets

| Dataset | Rocket | XGBoost | TST (GPU) |
| --- | --- | --- | --- |
| EthanolConcentration | 41.937 | 3.760 | 34.72 |
| FaceDetection | 279.033 | 57.832 | 67.8 |
| Handwriting | 6.705 | 1.836 | 134.4 |
| Heartbeat | 35.825 | 3.013 | 2.57 |
| InsectWingBeat | - | 64.883 | 4565 |
| JapaneseVowels | 5.032 | 0.527 | 4.71 |
| PEMS-SF | 369.198 | 150.879 | 341 |
| SelfRegulationSCP1 | 30.578 | 0.967 | 3.46 |
| SelfRegulationSCP2 | 28.286 | 1.213 | 97.3 |
| SpokenArabicDigits | 65.143 | 3.129 | 73.2 |
| UWaveGestureLibrary | 3.078 | 0.636 | 2.90 |

Table 7: Total training time (time until maximum accuracy is recorded) in seconds: for the fastest currently available methods (Rocket, XGBoost) on the same CPU, as well as for our fully supervised transformer models on a GPU. On a CPU, training for our model is typically at least an order of magnitude slower.

As regards deep learning models, LSTMs are well known to be slow, as they require $O(w)$ sequential operations (where $w$ is the length of the time series) for each sample, with the complexity per layer scaling as $O(N \cdot d^2)$, where $d$ is the internal representation dimension (hidden state size). We refer the reader to the original transformer paper (Vaswani et al., 2017) for a detailed discussion about how tranformers compare to Convolutional Neural Networks in terms of computational efficiency.

| Dataset Name | TST | SVR | Random Forest | XGBoost | 1-NN -ED | 5-NN -ED | 1-NN-DTWD | 5-NN-DTWD | Rocket | FCN | ResNet | Inception | Rel diff from 2nd best |
| --- | --- | --- | --- | --- | --- | --- | --- | --- | --- | --- | --- | --- | --- |
| AppliancesEnergy | 1 | 6 | 5 | 7 | 11 | 9 | 12 | 8 | 2 | 3 | 4 | 10 | -0.361 |
| BenzeneConcentration | 1 | 7 | 3 | 2 | 12 | 11 | 9 | 8 | 5 | 10 | 6 | 4 | -0.225 |
| BeijingPM10Quality | 1 | 7 | 3 | 2 | 12 | 9 | 11 | 8 | 10 | 4 | 5 | 6 | -0.067 |
| BeijingPM25Quality | 1 | 10 | 6 | 2 | 11 | 9 | 12 | 8 | 5 | 3 | 7 | 4 | -0.101 |
| LiveFuelMoistureContent | 2 | 3 | 5 | 4 | 12 | 7 | 11 | 6 | 1 | 8 | 10 | 9 | -0.038 |
| IEEEPPG | 2 | 10 | 5 | 4 | 7 | 3 | 12 | 8 | 11 | 9 | 6 | 1 | |

Table 8: Relative ranks of all methods on Regression datasets. Last column indicates relative diff of our method from the 2nd best, whenever the rank is 1.

| Problem | TST | Rocket | XGBoost | LSTM | Franseschi et al | DTW_D | Rel. Diff from 2nd best |
|---|---|---|---|---|---|---|---|
| EthanolConcentration | 4 | 1 | 2 | 3 | 6 | 5 | |
| FaceDetection | 1 | 2 | 3 | 4 | 6 | 5 | -0.041 |
| Handwriting | 3 | 1 | 5 | 6 | 2 | 4 | |
| Heartbeat | 1 | 2 | 4 | 5 | 3 | 6 | -0.02 |
| JapaneseVowels | 1 | 3 | 5 | 6 | 2 | 4 | -0.008 |
| PEMS-SF | 2 | 3 | 1 | 6 | 5 | 4 | |
| SelfRegulationSCP1 | 1 | 2 | 3 | 6 | 4 | 5 | -0.017 |
| SelfRegulationSCP2 | 1 | 4 | 5 | 6 | 2 | 3 | -0.055 |
| SpokenArabicDigits | 1 | 4 | 5 | 6 | 3 | 2 | -0.035 |
| UWaveGestureLibrary | 2 | 1 | 5 | 6 | 4 | 3 | |

Table 9: Relative ranks of methods on the Classification datasets. The last column is the relative diff of our method from the 2nd best, whenever we rank 1st.

| Dataset | Task (Metric) | Sep., Bern. | Sync., Bern. | Sep., Stateful | Sync., Stateful |
|---|---|---|---|---|---|
| Heartbeat | Classif. (Accuracy) | 0.761 | 0.756 | **0.776** | 0.751 |
| InsectWingbeat | Classif. (Accuracy) | 0.641 | 0.632 | **0.687** | **0.689** |
| SpokenArabicDigits | Classif. (Accuracy) | 0.994 | 0.994 | **0.998** | **0.996** |
| PEMS-SF | Classif. (Accuracy) | 0.873 | 0.879 | **0.896** | 0.879 |
| BenzeneConcentration | Regress. (RMSE) | 0.681 | **0.493** | 0.494 | 0.684 |
| BeijingPM25Quality | Regress. (RMSE) | 57.241 | 59.529 | **53.492** | 59.632 |
| LiveFuelMoistureContent | Regress. (RMSE) | 44.398 | 43.519 | **43.138** | 43.420 |

Table 10: Comparison of four different input value masking schemes evaluated for unsupervised learning on 4 classification and 3 regression datasets. Two of the variants involve separately generating the mask for each variable, and two involve a single distribution over "time steps", applied synchronously to all variables. Also, two of the variants involve sampling each "time step" independently based on a Bernoulli distribution with parameter $p = r = 15\%$, while the remaining two involve using a Markov chain with two states, "masked" or "unmasked", with different transition probabilities $p_m = \frac{1}{l_m}$ and $p_u = p_m \frac{r}{1-r}$, such that the masked sequences follow a geometric distribution with a mean length of $l_m = 3$ and each variable is masked on average by $r = 15\%$. We observe that our proposed scheme, separately masking each variable through stateful generation, performs consistently well and shows the overall best performance across all examined datasets.

| Dataset | Task (Metric) | LayerNorm | BatchNorm |
|---|---|---|---|
| Heartbeat | Classif. (Accuracy) | 0.741 | **0.776** |
| InsectWingbeat | Classif. (Accuracy) | 0.658 | **0.684** |
| SpokenArabicDigits | Classif. (Accuracy) | **0.993** | **0.993** |
| PEMS-SF | Classif. (Accuracy) | 0.832 | **0.919** |
| BenzeneConcentration | Regress. (RMSE) | 2.053 | **0.516** |
| BeijingPM25Quality | Regress. (RMSE) | 61.082 | **60.357** |
| LiveFuelMoistureContent | Regress. (RMSE) | 42.993 | **42.607** |

Table 11: Performance comparison between using layer normalization and batch normalization in our supervised transformer model. The batch size is 128.

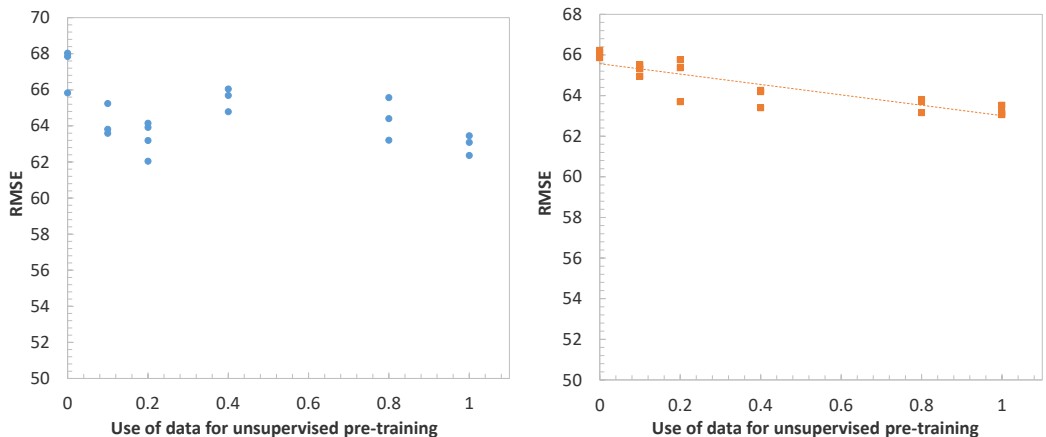

Figure 3: Root Mean Squared Error of a given model as a function of the number of samples (here, shown as a proportion of the total number of samples in the training set) used for unsupervised pre-training. Two levels of label availability (used for supervised learning) are depicted: 10% (left panel) and 20% (right panel) of all training data labels. Note that a horizontal axis value of 0 means fully supervised learning only, while all other values correspond to unsupervised pre-training followed by supervised fine-tuning.

| **Dataset** | **Task (Metric)** | **Static** | | **Fine-tuned** | |
|---|---|---|---|---|---|
| | | Metric | Epoch time (s) | Metric | Epoch time (s) |
| Heartbeat | Classif. (Accuracy) | 0.756 | **0.082** | **0.776** | 0.14 |
| InsectWingbeat | Classif. (Accuracy) | 0.236 | **4.52** | **0.687** | 6.21 |
| SpokenArabicDigits | Classif. (Accuracy) | 0.996 | **1.29** | **0.998** | 2.00 |
| PEMS-SF | Classif. (Accuracy) | 0.844 | **0.208** | **0.896** | 0.281 |
| BenzeneConcentration | Regress. (RMSE) | 4.684 | **0.697** | **0.494** | 1.101 |
| BeijingPM25Quality | Regress. (RMSE) | 65.608 | **1.91** | **53.492** | 2.68 |
| LiveFuelMoistureContent | Regress. (RMSE) | 48.724 | **1.696** | **43.138** | 3.57 |

Table 12: Performance comparison between allowing all layers of a pre-trained transformer to be fine-tuned, versus using static ("extracted") representations of the time series as input to the output layer (which is equivalent to freezing all model layers except for the output layer). The per-epoch training time on a GPU is also shown.

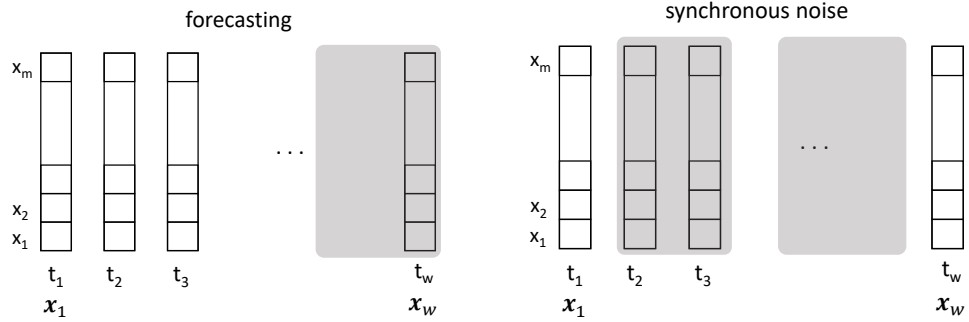

Figure 4: Masking schemes within our transformer encoder framework: for implementation of forecasting objective *(left)*, for an alternative unsupervised learning objective involving a single noise distribution over time steps, applied synchronously to all variables *(right)*.

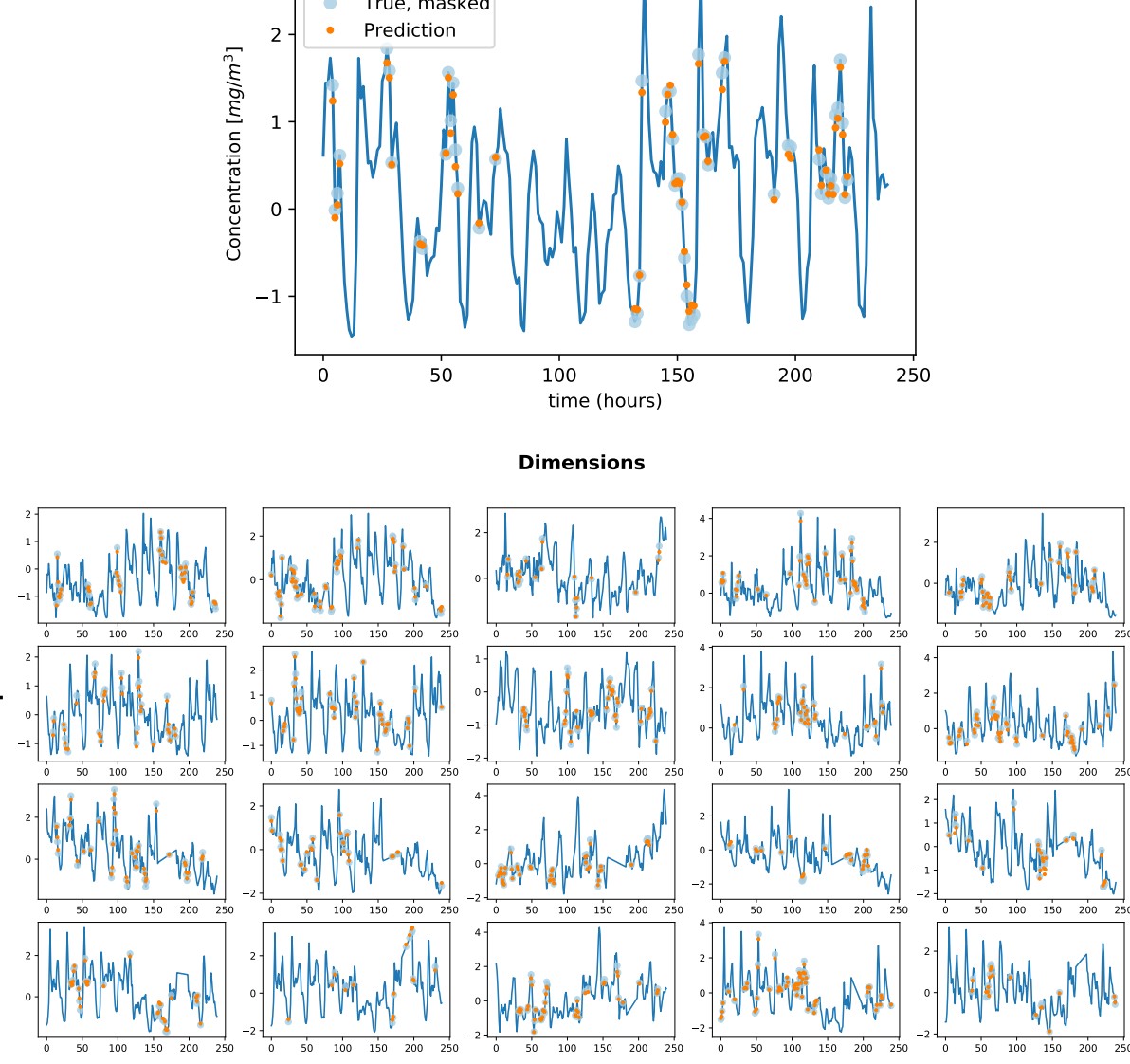

Figure 5: **Top:** Imputation of missing values in the test set of BenzeneConcentration dataset. The continuous blue line is the ground truth signal, the light blue circles indicate the values hidden from the model and the orange dots its prediction. We observe that imputed values approximate true values very well, even in cases of rapid transitions and in cases where many contiguous values are missing. **Bottom:** Same, shown for 5 different dimensions of the time series (here, these are concentrations of different substances) as columns and 4 different, randomly selected samples as rows.

## A.4 Advantages of transformers

Transformer models are based on a multi-headed attention mechanism that offers several key advantages and renders them particularly suitable for time series data:

- They can concurrently take into account long contexts of input sequence elements and learn to represent each sequence element by selectively attending to those input sequence elements which the model considers most relevant. They do so without position-dependent prior bias; this is to be contrasted with RNN-based models: a) even bi-directional RNNs treat elements in the middle of the input sequence differently from elements close to the two endpoints, and b) despite careful design, even LSTM (Long Short Term Memory) and GRU (Gated Recurrent Unit) networks practically only retain information from a limited number of time steps stored inside their hidden state (vanishing gradient problem (Hochreiter, 1998; Pascanu et al., 2013)), and thus the context used for representing each sequence element is inevitably local.

- Multiple attention heads can consider different representation subspaces, i.e., multiple aspects of relevance between input elements. For example, in the context of a signal with two frequency components, $1/T_1$ and $1/T_2$ , one attention head can attend to neighboring time points, while another one may attend to points spaced a period $T_1$ before the currently examined time point, a third to a period $T_2$ before, etc. This is to be contrasted with attention mechanisms in RNN models, which learn a single global aspect/mode of relevance between sequence elements.

- After each stage of contextual representation (i.e., transformer encoder layer), attention is redistributed over the sequence elements, taking into account progressively more abstract representations of the input elements as information flows from the input towards the output. By contrast, RNN models with attention use a single distribution of attention weights to extract a representation of the input, and most typically attend over a single layer of representation (hidden states).

## A.5 Hyperparameters

| Parameter | Value |
|---|---|
| activation | gelu |
| dropout | 0.1 |
| learning rate | 0.001 |
| pos. encoding | learnable |

Table 13: Common (fixed) hyperparameters used for all transformer models.

| Parameter | Value |
|---|---|
| dim. model | 128 |
| dim. feedforward | 256 |
| num. heads | 16 |
| num. encoder blocks | 3 |
| batch size | 128 |

Table 14: Hyperparameter configuration that performs reasonably well for all transformer models.

| Dataset | num. blocks | num. heads | dim. model | dim. feedforward |
|---|---|---|---|---|
| AppliancesEnergy | 3 | 8 | 128 | 512 |
| BenzeneConcentration | 3 | 8 | 128 | 256 |
| BeijingPM10Quality | 3 | 8 | 64 | 256 |
| BeijingPM25Quality | 3 | 8 | 64 (128) | 256 |
| LiveFuelMoistureContent | 3 | 8 | 64 | 256 |
| IEEEPPG | 3 | 8 | 512 | 512 |

Table 15: Supervised TST model hyperparameters for the multivariate regression datasets

| Dataset | num. blocks | num. heads | dim. model | dim. feedforward |
|---|---|---|---|---|
| AppliancesEnergy | 3 | 16 | 128 | 512 |
| BenzeneConcentration | 1 | 8 | 128 | 256 |
| BeijingPM10Quality | 3 | 8 | 64 | 256 |
| BeijingPM25Quality | 3 | 8 | 128 | 256 |
| LiveFuelMoistureContent | 3 | 8 | 64 | 256 |
| IEEEPPG | 4 | 16 | 512 | 512 |

Table 16: Unsupervised TST model hyperparameters for the multivariate regression datasets

| Dataset | num. blocks | num. heads | dim. model | dim. feedforward |
|---|---|---|---|---|
| EthanolConcentration | 1 | 8 | 64 | 256 |
| FaceDetection | 3 | 8 | 128 | 256 |
| Handwriting | 1 | 8 | 128 | 256 |
| Heartbeat | 1 | 8 | 64 | 256 |
| JapaneseVowels | 3 | 8 | 128 | 256 |
| PEMS-SF | 1 | 8 | 128 | 512 |
| SelfRegulationSCP1 | 3 | 8 | 128 | 256 |
| SelfRegulationSCP2 | 3 | 8 | 128 | 256 |
| SpokenArabicDigits | 3 | 8 | 64 | 256 |
| UWaveGestureLibrary | 3 | 16 | 256 | 256 |

Table 17: Supervised TST model hyperparameters for the multivariate classification datasets

| Dataset | num. blocks | num. heads | dim. model | dim. feedforward |
|---|---|---|---|---|
| EthanolConcentration | 1 | 8 | 64 | 256 |
| FaceDetection | 3 | 8 | 128 | 256 |
| Handwriting | 3 | 16 | 64 | 256 |
| Heartbeat | 1 | 8 | 64 | 256 |
| JapaneseVowels | 3 | 8 | 128 | 256 |
| PEMS-SF | 1 | 8 | 256 | 512 |
| SelfRegulationSCP1 | 3 | 16 | 256 | 512 |
| SelfRegulationSCP2 | 3 | 8 | 256 | 512 |
| SpokenArabicDigits | 3 | 8 | 64 | 256 |
| UWaveGestureLibrary | 3 | 16 | 256 | 512 |

Table 18: Unsupervised TST model hyperparameters for the multivariate classification datasets

