# OpenReview forum: "A Transformer-based Framework for Multivariate Time Series Representation Learning"
_ICLR.cc/2021/Conference — Reject_

### Official Review · AnonReviewer4 · 2020-10-27
**Representation learning on same dataset without transfer**

**Rating:** 4
**Confidence:** 4

**Review:**


# Summary:

The paper proposes an unsupervised learning framework, similar to BERT idea but for multivariate time series.
### Architecture:
- Inputs are projected into d-dimensional vector space where d is the dimension of the transformer model sequence element representations.
- A learned positional encoding is added to the input which is passed to the transformer encoder.
-The output of the encoder is then passed to another projection layer that depends on the application (i.e classification or regression)
-The transformer architecture is similar to Vaswani et al. (2017), however, they replaced layer norm with batch norm.

###  Unsupervised Learning Training:
For unsupervised learning, the paper proposed taking the input data masking it, and predicting an output sequence, the loss is then calculated as the mean square error between the actual input and the predicted input.

###  Evaluation and Results:
- The paper considered two tasks both regression and classification.  For regression, the proposed method was evaluated on datasets from the Monash University, UEA, UCR Time Series Regression Archive Tan et al. (2020a); for classification, the paper used the UEA Time Series Classification Archive (Bagnall et al., 2018).
- For each dataset, they showed the results of using the proposed transformer architecture in a supervised manner and then using the same dataset for pretraining the transformer by the proposed masking approach in an unsupervised manner followed by fine-tuning using labels. Both methods were compared to other state of the art methods for the same task.
- For multivariate regression, the pretraining followed by supervised training showed improvements in 3 out of 6 datasets. For classification 7 out of 11 datasets showed improvements.

# Strength:
- The paper focues  an important yet a relatively unexplored area.
- The paper is clear, well written and well motivated.
- The paper benchmarked accorss multiple datasets on both classification and regression tasks.

# Weakness:
- My main concern is the lack of novelty the paper is basically suggesting to use a  transformer encoder and add a dense layer before and after, and if we use unsupervised training of the transformer with the same dataset we ***may*** achieve better results.
- The main success of BERT is the ability to transfer and improve the performance on unretaleted task however here they did not include any experiments showing that if we train on model we can use to transfer on another model. I believe critical experiments that are missing is unsupervised training say with dataset 1 and the fine tuning on dataset 2 and showing improvements on multiple task (However, even if this experiments are provided I still believe the paper lacks novelty, maybe invistaginting what properties are being transferred in multivariate time series and showing difference between transferring from unvariate to multivariate and vice versa will help).
- The paper replaced layer norm with batch norm and only stated "here we instead use batch normalization, because it can mitigate the effect of outlier values in time series, an issue that does not arise in NLP word embedding" they did not show the effect of using batch norm on the accuracy or gave any insights on why it *mitigate the effect of outlier values in time series*.
- The code implementing the paper is not provided.
- The effect of changing values variable r in masking is not investigated.

---

> ### Author Response · Authors · 2020-11-18
> **Response to comments, Part 1: Novelty and main contributions, unsupervised learning**
>
> We thank the reviewer for their feedback, which we will use for improving the presentation of the paper and complementing it with additional experiments. Our response to the comments follows.
>
> “My main concern is the lack of novelty the paper is basically suggesting to use a transformer encoder and add a dense layer before and after, and if we use unsupervised training of the transformer with the same dataset we may achieve better results.”
>
> We consider our main contribution to be developing a general framework which allows the same model to be conveniently used (and fine-tuned/re-used) for different objectives: imputation, forecasting, classification and regression of multivariate time series, while at the same time allowing to effectively leverage unsupervised representation learning. This required suitably adapting a model architecture which has never before been used for the examined problems, similarly to other recently published work in deep learning  for this data modality (e.g., InceptionTime [Fawaz et al., 2019a] and ResNet [Fawaz et al., 2019b], which were adapted from models used for computer vision); however, we additionally consider the versatility/generality of our framework an important differentiator. Moreover, our proposed method represents an outstanding improvement on the state of the art in the context of time series classification and regression: it performs significantly better than the best currently available methods, while no other method, including the aforementioned ones, manages to meaningfully differentiate itself from the rest in terms of performance. Despite decades of innovation and meticulously engineered approaches on those problems, it is telling that the second best methods after ours are XGBoost, which does not even take sequence order into account, and ROCKET, which is based on a linear classifier on top of a flat collection of randomly initialized convolutional filters.
>
> We additionally believe that our method constitutes an important landmark, because through the use of our novel unsupervised objective it becomes the only method which has been shown to successfully leverage unlabeled data in order to improve performance and push the state of the art in time series regression and classification. Utilizing unlabeled time series data is immensely interesting for nearly all domains in the sciences and in industry. Interestingly, we show that it can accomplish this even when the number of available unlabeled data points is very limited, and that it can in fact benefit even from reusing labeled data samples through unsupervised learning.
> Nevertheless, we cannot guarantee that every single dataset will benefit from unsupervised learning, because datasets can display great variability in characteristics such as number of samples, data dimensionality, length of time series and overall difficulty of the task (which can be additionally affected by the nature of the relationship between input variables themselves or input variables and the target). For example, it is possible that enough samples exist such that a relatively “easy task” can be learned sufficiently well directly through unsupervised learning. Conversely, a very challenging task (where the target can hardly be predicted from the input variables), or a task that involves input variables which are completely independent from one another, will not benefit much from unsupervised learning, which aims at learning to model inter-dependencies between variables (besides learning to use past and future values of a given single variable).

---

> ### Author Response · Authors · 2020-11-18
> **Response to comments, Part 2: Evaluating potential for transfer learning**
>
> “The main success of BERT is the ability to transfer and improve the performance on unretaleted task however here they did not include any experiments showing that if we train on model we can use to transfer on another model. I believe critical experiments that are missing is unsupervised training say with dataset 1 and the fine tuning on dataset 2 and showing improvements on multiple task (However, even if this experiments are provided I still believe the paper lacks novelty, maybe invistaginting what properties are being transferred in multivariate time series and showing difference between transferring from unvariate to multivariate and vice versa will help).”
>
> It is important to note that the suggested critical experiment of training on some datasets and fine-tuning on different datasets makes sense in natural language processing, because in that field the transformer in any case encodes the **same entity, natural language** (i.e. the input consists of word vectors), and simply the *objective* changes between datasets (e.g. summarization, classification, question answering etc). However, in each of the examined datasets here, the time series  represent **completely different physical quantities** (molecule concentration, accelerometer readings, heart monitoring, power consumption etc), so the same encoder would be asked to interpret vectors of completely different physical meaning and dimensions. In other words, **across NLP datasets the nature of the data does not change, only the objective, while across the time series datasets the nature of the data itself changes**. Therefore, it becomes evident that in this case, transferability experiments only make sense again in terms of *objectives*, but within the same dataset, such as imputation, classification, regression, in the way that we have demonstrated in this work.
>
> [To further illustrate the above: there are only 2 cases which would allow “dataset transferability” experiments: there happen to be 2 pairs of datasets in the archives that indeed contain time series data of the same nature, with only the target changing from the one dataset of each pair to the other: BeijingPM25 - BeijingPM10, where we predict air quality from hourly concentration of pollutant molecules and atmospheric conditions, and SelfRegulationSCP1 - SelfRegulationSCP2, where we predict cursor movement based on 6-channel electroencephalogram (EEG) signals. Therefore, the performance we report for the pre-trained/fine-tuned models on these datasets can be technically seen as a "dataset transferability" experiment, since we are “using data for unsupervised learning from one dataset, and fine-tuning them on another”. However, each pair shares the exact same input data. Obviously, this is a trivial case and it is essentially not any different from the usual experiments we do on any other dataset (de-noising in an unsupervised way, and then fine-tuning for regression/classification on the same data), but this technicality is the only way to do transfer learning between different datasets in the archives that are currently available.]
>
> In a similar way as the BERT paper, the authors of which pretrain a model on using an unsupervised objective and then show performance gains of the **same model** for other tasks (classification, question answering etc), we have shown here how we can use data from a given dataset to pretrain on an unsupervised task, and then achieve better performance on an unrelated downstream task (classification, regression) by using weights from the pretrained model. Using the transformer encoder representations as inputs to different models is an interesting extension, but it was neither in the scope of BERT nor our present paper.  For any interested researcher, in our framework, this extension is possible either by precomputing the final representations, storing them and using them as inputs to an arbitrary model, or simply by freezing the transformer encoder layers and changing the final output layer from linear to any other model that can be represented by a computation graph.

---

> ### Author Response · Authors · 2020-11-18
> **Response to comments, Part 3: BatchNorm, releasing code, masking ratio**
>
> “The paper replaced layer norm with batch norm and only stated "here we instead use batch normalization, because it can mitigate the effect of outlier values in time series, an issue that does not arise in NLP word embedding" they did not show the effect of using batch norm on the accuracy or gave any insights on why it mitigate the effect of outlier values in time series.”
>
> Following the reviewer’s suggestion, we will report differences in performance when using batch normalization instead of layer normalization.
> As we mention in our paper, we replace layer normalization, which in general “cheaply” approximates the effect of batch normalization, because it seems that the reason why it is empirically preferred to batch normalization in NLP is related to the special statistics of NLP data  (Shen et al., 2020).
> An intuition as to why batch normalization can help with outlier values in time series is the following: time series data often contain outlier values within certain samples, or even entire “outlier samples” (e.g. a sample where one sensor was always inactive). Using statistics from an entire batch of samples, batch normalization performs an affine transform (shift and rescaling) on activations corresponding to outlier samples within a batch in a way that does not allow them to significantly affect the output and thus the learned model parameters. They therefore have a regularizing effect, reducing the potential of such outliers to lead to overfitting. (Equivalently, batch normalization can be seen as stabilizing the activation distributions before the input of succeeding layers.) As a result of this regularization, BatchNorm has been shown to have an important effect on the training process (Santurkar et al., NeurIPS 2018): it makes the optimization landscape significantly smoother, which induces a more stable behavior of the gradients, allowing for faster training.
>
>
> “The code implementing the paper is not provided.”
>
> Our plan is to publicly release our code in an online repository as soon as possible. Unfortunately, we are unable to do so for the time being due to Intellectual Property issues raised by an involved industry partner. Authorization may require months.
>
>
> “The effect of changing values variable r in masking is not investigated”
>
> We did not observe a systematic improvement of performance when trying other values of $r$, so we kept our initial, intuitive guess of $r=0.15$, which was inspired from BERT, where 15% of input tokens are masked. We may more carefully optimize this value in the future, however we do not expect performance to improve substantially.

---

### Official Review · AnonReviewer3 · 2020-10-28
**Interesting adaptation of BERT objective to timeseries self-supervised training but performance and design decisions not rigorously evaluated.**

**Rating:** 4
**Confidence:** 3

**Review:**

This paper uses transformer to improve mutlivariate time series classification and regression using a BERT inspired self-supervised loss. The authors show improvement over multiple standard datasets. The use of self-supervision improves performance in lower (labeled) data regime.


#### Strong points

- The method is well explained and take good inspiration of BERT pretraining.
- The data-available experiment shows that training with self-supervision helps for classification.
- This work is well positioned in the literature of the field.

#### Weak points

- The timeserie representation is obtained by concatenation of the tokens' representations. This means that the representations are not of fixed size (or contain padding representations).
- The introduced masking loss could suffer from discrepancy between training and inference where all the covariates at a given timesteps need to be predicted. Moreover, the model could rely on very correlated covariates (which frequently happens) to recover the masked variables limiting the effect of self-supervised training. I would like to see if masking all variables at some times step helps/hurts the pretraining.
- The paper does not consider if finetuning is needed or if the network could be frozen and only a small MLP could be learned on top of the "pre-extracted times-series representations".
- The hyperparameters of the network are chosen per dataset but are not reported. It is not clear if the tuning is done with cross validation or looking at the test error. The authors should report the performance of the transformer with common hyperparameters specified in the Appendix. This would be more fair to compare with Franceschi et al., who do not tune the hyperparameters of their encoder.
- The claim in the abstract that the method offers "computational efficiency" is not backed with evidence. Section A.4 "Execution Time" does not report execution time of the method compared to the baselines.


#### Decision

I tend to reject this paper. The idea is interesting but some design decisions of the method (representation pooling, self-supervised loss, need for finetuning) should be better justified. Secondly the evaluation of the´ method should be more precise (hyperparameters tuning, speed).

#### Questions/Remarks

- In section 3.1 "because the computational complexity and the number of parameters of the model
scale as $O(w^2)$ with the input sequence length $w$". The number of parameters of the transformer architecture does not scale with the input length. Only the memory footprint and the computation scales quadratically.

- "we note that similar to the case of word embeddings, the positional encodings generally appear not to interfere with the numerical information of the time series". Do you have evidence for this?

- Have you tried PowerNorm by Shen et al. (2020) that you cite in 3.1?

- You could mention Temporal Fusion Transformers for Interpretable Multi-horizon Time Series Forecasting (Lim et al., 2019) arXiv:1912.09363 for another use of transformers for univariate (quantile) forecasting.

- Add the dataset you are experimenting with in the caption of Figure 2.

#### Additional feedback

- Figure 2: chose different colors for left and right figures.
- Table 1: consider using booktabs for table format and transpose dataset and models to make it fit in the margin.
- Section 3.2: The sentence starting with "The reason ..." that spans 6 lines could be split and reworded.

---

> ### Author Response · Authors · 2020-11-18
> **Response to comments, Part 1: time series representation**
>
> We thank the reviewer for the thoughtful feedback, questions and comments, which give us the opportunity to explain our approach more thoroughly and to improve the paper in terms of phrasing, presentation and experiments [updated version will be uploaded soon]. Our response follows:
>
> “The timeserie representation is obtained by concatenation of the tokens' representations. This means that the representations are not of fixed size (or contain padding representations).”
>
> [We first note that, when computing the output, the representations corresponding to padding are zeroed-out and do not contribute to the computation.] We do not advocate using the concatenation of representations as a general way of representing the time series: the concatenation was only used for the particular objectives evaluated, i.e. regression and classification, and is not meant for extraction and storage as a universal representation of the time series. As we note in A.1 of the appendix, for other purposes, the individual representations at each time step can be used separately (i.e. as done for imputation) or combined in other ways (e.g. weighted mean over some or all time steps), depending on the application. We consider this flexibility an advantage of our approach: for example, if we wanted to compare time series with one another only based on their beginning/ending, we could selectively choose to average the encoder output embeddings corresponding to the first/last few time steps.
>
> In future work, we plan to explore which types of representations work best for each application, e.g. addressing retrieval, clustering, anomaly detection etc. However, contrary to Franseschi et al., whose goal is to extract and store *universal representations of general utility*, our (less ambitious) goal is to *use the appropriate representations for each intended objective*. This goal is guided by the observed shortcomings of universal representations, e.g. in fields such as medical time series (Lyu et al, NeurIPS ML4Health Workshop 2018), as well as insights on transformer representations in the field of natural language processing: in practice, researchers fine-tune pre-trained transformer models instead of extracting and storing representations, e.g. BERT, T5 (Raffel et al, JMLR 2020).

---

> ### Author Response · Authors · 2020-11-18
> **Response to comments, Part 2: Masking scheme, fine-tuning vs pre-computed representations**
>
> “The introduced masking loss could suffer from discrepancy between training and inference where all the covariates at a given timesteps need to be predicted. Moreover, the model could rely on very correlated covariates (which frequently happens) to recover the masked variables limiting the effect of self-supervised training. I would like to see if masking all variables at some times step helps/hurts the pretraining.”
>
> In case the objective during inference would necessitate predicting all covariates at given timesteps, there would indeed be a discrepancy; in this case, we would certainly suggest using a mask that includes instances of masking the entire “input vector” (all variables) at some time steps. More generally, using our framework allows to conveniently use the proper mask for the proper task. Having said that, predicting all covariates at once was not part of any objective within the frame of the present work. However, we completely agree with the reviewer that it is beneficial to evaluate different masking schemes with respect to their performance in the examined tasks of classification and regression (we clarify that in the context of this work and the literature we refer to, regression means predicting a single scalar value using the entire multivariate time series as input, e.g. predicting heart rate using time-aligned ECG signals from several sensors); therefore, we will report results of this investigation in the final version of our manuscript.
> Finally, we note that the approach we have chosen has some theoretical advantages: we wish to encourage the model to learn co-dependencies between variables, and our masking scheme ensures that a wide variety of different variable combinations will be seen during training. In certain datasets, the data is merely *represented* as time series and included in the time series archives, however the variables are not in fact signals evolving with time, and “past” values don’t necessarily help predicting “future” values - using spectroscopic data (absorption of light at each frequency) to predict a substance’s concentration is such an example: absorption at each frequency is characteristic for each chemical substance, and cannot be predicted by absorption at preceding or subsequent frequencies (at least remote ones). In principle, there is no way of knowing the missing values when masking all variables (absorption spectra) for a continuous band of frequencies of a certain width (note that more than 1-2 “time steps”/frequency values have to be masked, otherwise the prediction task becomes trivial by using immediately preceding-succeeding values).
>
> “The paper does not consider if finetuning is needed or if the network could be frozen and only a small MLP could be learned on top of the "pre-extracted times-series representations".”
>
> Following the reviewer’s suggestion, we will report the effect of fine-tuning the entire model vs training only the output layer/MLP and freezing preceding layers (equivalently, extracting and using immediately preceding activations as an input).
> Nevertheless, we wish to clarify that our goal differs from the goal of e.g. Franseschi et al., which is to extract global representations of universal utility regardless of downstream task. While such universal/static representations indeed save computation, and they may also have other uses, we consider that, at least in our case, and for the objectives we address (classification, regression, imputation, forecasting), they are not practical, since the size of the final representations can be larger than the input vector size. Extracted representations occupy storage space on disk; also, when they are needed for computation, they have to first be loaded on CPU and GPU memory, which in practice diminishes the computational benefits of this approach. Importantly, given the opportunity to train on a dataset, scientists/engineers would always prefer to optimize representations to perform best for their task of interest. We are thus aligned with the current paradigm of NLP transformers, according to which researchers fine-tune pre-trained models on new tasks, instead of using stored static representations. At the same time, in the case of our models, allowing the encoder weights to be fine-tuned alongside the output layer while training for classification/regression incurs a negligible overhead, and fine-tuning on most of the examined datasets requires much less than an hour overall (pre-trained models also converge faster than models trained from scratch). Thus, in our view, it is not justified to compromise model performance for a negligible computational speedup; however, we agree that it is useful to quantify what the speedup - performance trade-off actually is, and we plan to report it.

---

> ### Author Response · Authors · 2020-11-18
> **Response to comments, Part 3: Hyperparameters, computational efficiency**
>
> “The hyperparameters of the network are chosen per dataset but are not reported. It is not clear if the tuning is done with cross validation or looking at the test error. The authors should report the performance of the transformer with common hyperparameters specified in the Appendix. This would be more fair to compare with Franceschi et al., who do not tune the hyperparameters of their encoder.”
>
> To select hyperparameters, for each dataset we randomly split the training set (predefined by the archive) in two parts, 80%-20%, and used the 20% as a validation set for hyperparameter tuning. After fixing the hyperparameters, the entire training set was used to train the model again, which was finally evaluated on the test set (also defined by the archive). Interestingly, we found that the chosen hyperparameters were not sensitive to the random set used to tune them; more generally, the same set of hyperparameters (Table 6 in the Appendix) worked well in every case. However, as expected, because different datasets in the archive contain data with completely different characteristics, number of samples and physical nature, the optimal value of some hyperparameters (e.g. model dimension) did depend on the dataset. Our initial intention was to include hyperparameters as part of configuration files inside our repository - however, we can expand Table 6 to report the set of hyperparameters used for each of the 17 datasets that were used to evaluate our models.
>
> Franseschi et al. do not have a “supervised version” of their models and choose hyperparameters by only considering performance on their unsupervised task. They do not consider whether their end-to-end model performance, after unsupervised learning, is better than a purely supervised model using the same architecture (CNN with dilations, followed by dense layer).  We explicitly address the important question whether unsupervised learning can give an advantage over purely supervised models. To do this comparison fairly, we have to pick the optimal parameters for the supervised/pre-trained versions of the models - for example, it would be unfair to train a large model through many epochs of unsupervised training and fine-tune on only 300 available labeled samples, and compare it with the same exact model trained directly on these 300 labeled samples, because in practice, if one wanted to eschew unsupervised learning, they would pick an appropriate (smaller) model for unsupervised learning.
>
> The goal of Franseschi et al. is to extract global representations of universal utility. While there might exist some uses of such representations, in our view, this is not a realistic scenario of using datasets in practice, at least for the examined objectives of classification and regression. We do not see a reason why to use universal hyperparameters for all distinct datasets, given that each one of them consists of data of completely different physical nature (e.g. power consumption, accelerometer readings, health monitoring, absorption spectra etc) and very different characteristics; the only reason would be convenience in evaluating models on many different datasets, which only affects ML method researchers such as ourselves and Franseschi et al. In reality, given the opportunity to train on their datasets of interest, users/researchers would always prefer to optimize hyperparameters on their particular problem at hand. Our model and setup allow training on most of the examined datasets in a time much shorter than an hour, so customizing hyperparameters is certainly feasible.
>
> Nevertheless, despite the relatively minor dependence of hyperparameters on the dataset, our models’ large performance advantage over Franseschi et al. (they are better on 10 out of 11 datasets, achieving an average rank of 1.7 vs 3.7, and achieve an average relative improvement of approx. 37% in accuracy) still exists even if we use sub-optimal parameters (we can display such results as supplementary material); as an indication, we note that also our models which were trained in a purely supervised way outperform the models of Franseschi et al., to approximately the same extent as the ones which were pre-trained through the unsupervised objective and then fine-tuned, despite having undergone a different training process and generally having different hyperparameters.

---

> ### Author Response · Authors · 2020-11-18
> **Response to comments, Part 4: Number of parameters, positional encodings, miscellaneous**
>
> R: “In section 3.1 "because the computational complexity and the number of parameters of the model scale as $O(w^2)$ with the input sequence length ". The number of parameters of the transformer architecture does not scale with $w$ the input length. Only the memory footprint and the computation scales quadratically.”
>
> We thank the reviewer for pointing out this error. Our models’ architecture does include components whose number of parameters scale with the input length $w$, namely: the learnable positional embedding matrix, the output layer, and the batch normalization layers. However, unlike in the case of computational complexity, this dependence is linear and not quadratic, and we will change the formulation of the paper to reflect this.
>
> R: “"we note that similar to the case of word embeddings, the positional encodings generally appear not to interfere with the numerical information of the time series". Do you have evidence for this?”
>
> This statement was meant as a general observation based on performance: since our transformer models perform very well on the tasks they were evaluated on, significantly outperforming the state-of-the-art baselines, we conclude that the added positional embeddings do not catastrophically interfere with the numerical information of the time series, an observation which, at least to us, was not obvious in advance. Based on the performance of our models (they tend to perform better for higher dimensional input), we have some indication that the higher dimensional the input, the easier it is for the positional embeddings to be learned in a way so as to occupy a different subspace than the one in which the projected time series samples reside (and thus interfere less). However, isolating the effect of the positional embeddings is challenging - even if we concatenate them with the input vector instead of adding them, it will not be clear whether the difference in performance can be attributed to the overall change of dimensionality of the input. Nevertheless, we will change the phrasing to: “the positional encodings generally appear not to *significantly/catastrophically* interfere with the numerical information of the time series”, which makes the statement less categorical.
>
> R: "Have you tried PowerNorm by Shen et al. (2020) that you cite in 3.1?"
>
> Because PowerNorm was explicitly designed to deal with the statistics of NLP data, and its effectiveness was also only evaluated on NLP datasets, we didn’t prioritize experimenting with PowerNorm in the frame of this work; however, we would be interested in evaluating it in future work.
>
>
> R: “You could mention Temporal Fusion Transformers for Interpretable Multi-horizon Time Series Forecasting (Lim et al., 2019) arXiv:1912.09363 for another use of transformers for univariate (quantile) forecasting.”
>
> We thank the reviewer for pointing us to this interesting work, which we will add to the paragraph of related work on univariate forecasting.

---

> ### Comment · AnonReviewer3 · 2020-11-24
> **Please upload a revision**
>
> I would like to thank the authors for their answers. They addressed some of my concerns however I cannot judge the promised changes if they are not uploaded in the revision of the manuscript.

---

### Official Review · AnonReviewer1 · 2020-10-28
**a transformer is used for multivariate time series representation learning, but there are some unclear points about the proposed model and experimental settings.**

**Rating:** 4
**Confidence:** 5

**Review:**

This paper aims to develop a transformer-based pre-trained model for multivariate time series representation learning. Specifically, the transformer’s encoder is only used and a time-series imputation task is constructed as their unsupervised learning objective. This is a bit similar to the BERT model in NLP. But authors added a mask for each variable of the time series. After pretraining with this imputation loss, the transformer can be used for downstream tasks, such as regression and classification. As the authors mentioned on page 6, this is achieved by further fine-tuning all weights of the pre-trained transformer.

It is natural to use the transformer model from NLP for time series modeling since both sentences and time series are sequential data. In this work, the authors’ contributions or changes should include two-points: 1) constructing that imputation task for multivariate time series data. 2) using a learned positional encoding (page 4). I think these two things seem a bit interesting.
My concerns mainly include:

1. Actually, there are existing works that have tried to use the transformer for time series. But you didn’t compare them in your experiments. At least, I think you should clarify what’s your advantages comparing to these existing works:
- 2019 NeurIPS Enhancing the Locality and Breaking the Memory Bottleneck of Transformer on Time Series Forecasting
- 2020 CVPR Sketch-BERT: Learning Sketch Bidirectional Encoder Representation From Transformers by self-supervised Learning of Sketch Gestalt

3. As a core topic in this work, I encourage authors to clarify the definition of the time series representation. What’s a good representation in time series? There is a related work for time series representation learning:
- 2019 NeurIPS Unsupervised Scalable Representation Learning for Multivariate Time Series
I notice that they also train a dilated CNN model to get a single feature for a segment of time series. But in your case, you directly concatenate states from all-time steps to form your final vector. I just wonder if your strategy is reasonable. Because when we deal with long time series, concatenating them will create another long time-series/hidden state sequence. This may not be representation learning and it cannot handle the segment-level tasks such as classification. Thus, I encourage authors to give more insights into the questions what’s a good representation of the time series.

3. Besides, in the above NeurIPS representation learning paper, I find they didn’t fine-tune their model’s parameters. They just added SVM to test the performance of learned representations. But in this work, the authors fine-tuning their model. Thus, I wonder what’s the performance of your model without additional fine-tuning. What's your parameter settings for the fine-tuning procedure.

4. Another question I am concerned about is your learnable position encoding. It is with a shape of w-by-d where w should be your window length and d is your input dimension. Since time series is dynamic data and its length would be much longer, this design maybe not good as it largely increases the number of parameters.

5. For experimental results, authors mainly consider their model’s performance on regression and classification tasks. But I would like to see more analysis of their learned presentations. Adding more visualization analysis would be helpful to demonstrate your framework’s effectiveness.

6. Experimental settings are unclear. For example, since your datasets only contain train/test sets. How to pick your hyperparameters? Is it based on that performance on that test set? What kind of regression task is used in your regression experiments? Is it a one-step-ahead prediction task?

7. In Table 5, you claim your model is faster than the ROCKET model. But the running time reported for your model is per-epoch training time, not the total training time. I think this seems a bit unreasonable. Please check it.

---

> ### Author Response · Authors · 2020-11-17
> **Response to comments, Part 1: Existing work, definition of regression task**
>
> We thank the reviewer for the interesting questions and comments, which give us the opportunity to more thoroughly explain our approach and to improve the paper [updated version will be uploaded soon]. Our response follows:
>
> “1. Actually, there are existing works that have tried to use the transformer for time series. But you didn’t compare them in your experiments. At least, I think you should clarify what’s your advantages comparing to these existing works:
> 2019 NeurIPS Enhancing the Locality and Breaking the Memory Bottleneck of Transformer on Time Series Forecasting
> 2020 CVPR Sketch-BERT: Learning Sketch Bidirectional Encoder Representation From Transformers by self-supervised Learning of Sketch Gestalt”
>
> Regarding “2019 NeurIPS Enhancing the Locality and Breaking the Memory Bottleneck of Transformer on Time Series Forecasting”:
>
> This work (Li et al.), which we cite in the related work section, employs a _full transformer encoder-decoder_ architecture (as the original by Vaswani et al) for the purpose of *forecasting* in univariate time series. We could of course extend this method, in a similar way as our model, to work for multivariate instead of univariate time series, so this was not the main limitation. The main reason why it is not included in our comparison is that, although the transformer decoder is well suited for the generative task of subsequent values prediction and as such it is used for forecasting, it is not suitable for sequence classification and regression, the tasks our model and all other baselines are evaluated on. This is not simply a problem of parameter parsimony (although the additional decoder would indeed necessitate roughly as many parameters as the encoder): the decoder requires a _target sequence as input which differs from the input of the encoder_ and constitutes what the decoder should generate, it employs a decoder-encoder attention scheme where target representations are compared to the encoder’s input representations, and it generates a _sequence of successive output values_. This objective is very different from our objectives of classification and regression. Note that, having classification in mind, the BERT paper similarly dispensed with the decoder.
>
> At this point we take the opportunity to clarify what *regression* means in the specific context of our work and the time series literature we refer to, because we realize that this was unfortunately not very clear in our original manuscript. We thus address the question raised by the reviewer in their point 6: “What kind of regression task is used in your regression experiments? Is it a one-step-ahead prediction task?”. Regression here means predicting a single numeric value for a given sequence (time series sample). This numeric value is of a different nature than the numerical data appearing in the time series: for example, given a sequence of simultaneous measurements of two-channel PPG (photoplethysmogram) signals, three-axis acceleration signals, and one-channel ECG (electrocardiogram) signals, the regression task is to predict the heart rate; given a sequence of simultaneous temperature and humidity measurements of 9 rooms in a house, as well as weather and climate data such as temperature, pressure, humidity, wind speed, visibility and dewpoint, predict the total energy consumption in kWh of a house for that day. These scalar values are the numerical “labels” provided by the datasets to train and evaluate models. In the general case, one may want to predict multiple scalars (or a vector) instead of a single scalar; this corresponds to the dimensionality $n$ entering the equations in Section 3.1.
>
> It is important to note here that the goal of our work is to develop a general framework for time series representation learning, which easily facilitates both unsupervised learning as well as several downstream objectives. As such, besides unsupervised pre-training, our method can perform imputation, regression, classification as well as forecasting (the latter is implemented by modifying the objective mask, as outlined in the Appendix A.1 and Figure 4). Instead, the model by Li et al. can only perform forecasting. We do not evaluate our method on forecasting datasets in the scope of this work, but we will certainly compare the two methods in future work on forecasting, because it would be interesting to assess the effect of the decoder.
>
> We also note that the method of Li et al. includes a “log-sparse” attention mechanism, which is employed to mitigate the $O(w^2)$ dependence of computational complexity and memory of self-attention on the input sequence length $w$, at the possible expense of predictive performance. This is useful for their intended purpose of forecasting in their datasets, because they have to consider very long input sequences; by contrast, the datasets in the archives we have considered can be handled by the full attention mechanism, and thus we do not need to employ such a modification.

---

> ### Author Response · Authors · 2020-11-17
> **Response to comments, Part 2: Existing work**
>
> Regarding “2020 CVPR Sketch-BERT: Learning Sketch Bidirectional Encoder Representation From Transformers by self-supervised Learning of Sketch Gestalt”:
>
> We thank the reviewer for pointing us to this interesting work, which we were unaware of. The goal of the 2020 CVPR work is to learn representations of sketch drawings, using insights from the Gestalt theory of visual perception (principles of perceptual grouping). Although the authors, similarly to us, use a transformer encoder, their model takes an input of a very specific meaning, format and dimensionality: (∆x, ∆y, p1, p2, p3), where ∆x and ∆y are the values of *relative offsets* between current point in the drawing and previous point; (p1, p2, p3) is a *one-hot vector* indicating the state of each point, and p2 = 1 indicates the ending of one stroke, p3 = 1 means the ending of the whole sketch, and p1 = 1 represents the other sequential points of sketches. Likewise, most contributions of that paper are *very specific to their objective and data format*, e.g. learning stroke embeddings, or predicting separately the sketch point offsets (∆x,∆y) or state (p1, p2, p3) as an unsupervised objective. Therefore, unlike our model and all baselines we consider, that model cannot be used on multivariate time series data of arbitrary dimensionality and physical meaning and thus cannot participate in the comparison. By contrast, our model, which is more general, *can* be used for classification of sketch drawings using the same input representation as the 2020 CVPR paper, if one was interested in this problem; of course, we don’t necessarily expect it to perform that task as well as the model of the 2020 CVPR paper, which was specifically designed with that purpose in mind. As a side note, since classification was the objective of the 2020 CVPR paper, it also dropped the decoder, similar to us and unlike the Li et al. forecasting paper mentioned above.

---

> ### Author Response · Authors · 2020-11-17
> **Response to comments, Part 3: What constitutes a good representation**
>
> “As a core topic in this work, I encourage authors to clarify the definition of the time series representation. What’s a good representation in time series? There is a related work for time series representation learning:
> 2019 NeurIPS Unsupervised Scalable Representation Learning for Multivariate Time Series I notice that they also train a dilated CNN model to get a single feature for a segment of time series. But in your case, you directly concatenate states from all-time steps to form your final vector. I just wonder if your strategy is reasonable. Because when we deal with long time series, concatenating them will create another long time-series/hidden state sequence. This may not be representation learning and it cannot handle the segment-level tasks such as classification. Thus, I encourage authors to give more insights into the questions what’s a good representation of the time series.”
>
> Our models perform outstandingly better than all other baselines for regression and classification, and especially the models in the 2019 NeurIPS paper (Franseschi et al.) with respect to classification (Franseschi et al. did not consider regression): they are better on 10 out of 11 datasets, achieving an average rank of 1.7 vs 3.7, and achieve an average relative improvement of approx. 37% in accuracy compared to the models of Frenseschi et al. We are therefore very confident that the learned representations at the output of the transformer encoder are meaningful and encapsulate important information about the time series. However, *we do not advocate using the concatenation of representations as a general way of representing the time series*. The concatenation was only used for the particular objectives evaluated, i.e. regression and classification, and is not meant for extraction and storage as "*the* representation" of the time series. As we note in A.1 of the appendix, for other purposes, the individual representations at each time step can be used separately (e.g. as done for imputation) or combined in other ways (e.g. weighted mean over some or all time steps), depending on the application. We consider this flexibility an advantage of our approach: for example, if we wanted to compare time series with one another only based on their beginning/ending, we could selectively choose to average the encoder output embeddings corresponding only to the first/last few time steps.
>
> We completely agree with the reviewer on the importance of considering what constitutes a good representation, and exactly for this reason we plan to explore which types of representations work best for each application in future work, e.g. addressing retrieval, clustering, anomaly detection etc. However, contrary to Franseschi et al., whose goal is to extract and store *universal representations of general utility*, our (less ambitious) goal is to *use the appropriate representations for each intended objective*. This goal is guided by the observed shortcomings of universal representations, e.g. in fields such as medical time series (Lyu et al, NeurIPS ML4Health Workshop 2018), as well as insights on transformer representations in the field of natural language processing: in practice, researchers fine-tune pre-trained transformer models instead of extracting and storing representations, e.g. BERT, T5 (Raffel et al, JMLR 2020).

---

> ### Author Response · Authors · 2020-11-17
> **Response to comments, Part 4: Fine-tuning vs static representations, learnable positional encoding**
>
> “3. Besides, in the above NeurIPS representation learning paper, I find they didn’t fine-tune their model’s parameters. They just added SVM to test the performance of learned representations. But in this work, the authors fine-tuning their model. Thus, I wonder what’s the performance of your model without additional fine-tuning. What's your parameter settings for the fine-tuning procedure.”
>
> As noted above, the goal of Franseschi et al. is to extract global representations of universal utility, regardless of downstream task. While doing so indeed saves computation *for someone who intends to perform many different tasks*, and there might exist some other uses for such representations, in our view, and at least for the objectives we address (classification, regression, imputation, forecasting), this is not how datasets are used in a realistic scenario. Given the opportunity to train on a dataset, scientists/engineers would always prefer to optimize representations to perform best for their task of interest. We are thus aligned with the current paradigm of NLP transformers: researchers fine-tune pre-trained models on new tasks, instead of using stored static representations. In the case of our models, allowing the encoder weights to be fine-tuned alongside the output layer while training for classification/regression incurs a negligible overhead, and fine-tuning on most of the examined datasets requires much less than an hour overall (pre-trained models also converge faster than models trained from scratch). Thus, in our view, it is not justified to compromise model performance for a negligible computational speedup, which only grants a minor convenience in case one wants to perform many different tasks with the same data (something that mostly conveniences ML method researchers such as Franseschi et al. and us). Finally, an additional disadvantage of static representations is that they occupy storage space on disk; also, when they are needed for computation, they have to be loaded on CPU and GPU memory, which in fact diminishes the computational benefits of this approach. Nevertheless, for completeness, we intend to show the impact of fine-tuning in experiments which we will include in our final manuscript version. The hyperparameters of our models during fine-tuning stay the same as during pre-training; only the objectives (output layer and loss functions) change.
>
> “4. Another question I am concerned about is your learnable position encoding. It is with a shape of w-by-d where w should be your window length and d is your input dimension. Since time series is dynamic data and its length would be much longer, this design maybe not good as it largely increases the number of parameters.”
>
> Interestingly, our learnable positional encoding scheme is very efficient: it works just like learnable word embeddings in NLP, where the model needs to learn 1 vector for each word in the vocabulary of size $V$ (in the order of $10^4$ or $10^5$). Here, however, we avoid the costly embedding look-up operation (matrix multiplication), because while in NLP word indices can come in any order, e.g. $[12, 994, 3, 1763]$, the position indices always have the same order: $[0, 1, 2, … w-1]$. Thus, instead of matrix multiplication and addition, we here only have addition. Most importantly, in terms of the number of parameters, our approach is much more parsimonious than word embedding learning in NLP: this is because instead of $V$, the model only has to learn at maximum $w$ different positional vectors, where $w$ is the number of positions (i.e. the maximum length of the input window, e.g. $w = 512$). For each input sequence of length $L \leq w$ (longer sequences are trimmed, so $w$ would be determined such that we avoid trimming a significant number of samples), only $L$ positional encodings are used and updated by the model - the rest are completely ignored by the attention and have no effect in the forward pass or backward gradient calculation. However, if there is a need, we can see that $w$ could grow to similar sizes as $V$ in NLP, without any extra concerns about the number of parameters in this layer.

---

> ### Author Response · Authors · 2020-11-17
> **Response to comments, Part 5: Analysis of representations, tuning hyperparameters**
>
> “5. For experimental results, authors mainly consider their model’s performance on regression and classification tasks. But I would like to see more analysis of their learned presentations. Adding more visualization analysis would be helpful to demonstrate your framework’s effectiveness.”
>
>  We certainly share the reviewer’s interest in analyzing learned representations, and for this reason we plan to thoroughly explore representations in the context of several applications in future work, e.g. addressing retrieval, clustering, anomaly detection etc, where visualization is indeed very important for an intuitive understanding. In the scope of the present submission, our goal was to define our framework for supervised and unsupervised learning and showcase its potential on two problems which are widely considered important and difficult, and warrant publishing on their own right (most methods we compare with indeed were published as specific solutions to these problems). In this context, we note that:
>
> - The proposed method represents an outstanding improvement on the state of the art in the context of time series classification and regression: it performs significantly better than the best currently available methods, while no other method manages to meaningfully differentiate itself from the rest. Despite decades of innovation and meticulously engineered approaches on those problems, it is telling that the second best methods are XGBoost, which does not even take sequence order into account, and ROCKET, which is based on a linear classifier on top of a flat collection of randomly initialized convolutional filters.
> - Our method is the only method which has been shown to successfully leverage unlabeled data in order to improve performance and push the state of the art in time series regression and classification. Leveraging unlabeled time series data is immensely interesting for nearly all domains in the sciences and in industry. Importantly, we show that it can accomplish this even when the number of available unlabeled data points is very limited, and that it can in fact benefit even from reusing labeled data samples through unsupervised learning.
> - This performance is accomplished by 1) suitably adapting a model architecture which has never before been used for these problems, 2) employing a novel unsupervised objective which allows unsupervised representation learning, 3) developing a framework which allows the same model to be used for different objectives (imputation, forecasting, classification, regression).
>
> “6. Experimental settings are unclear. For example, since your datasets only contain train/test sets. How to pick your hyperparameters? Is it based on that performance on that test set?”
>
> To select hyperparameters, for each dataset we randomly split the training set (predefined by the archive) in two parts, 80%-20%, and used the 20% as a validation set for hyperparameter tuning. After fixing the hyperparameters, the entire training set was used to train the model again, which was finally evaluated on the test set (also defined by the archive). Interestingly, we found that the chosen hyperparameters were not sensitive to the random set used to tune them; more generally, the same set of hyperparameters (Table 6 in the Appendix) worked well in every case. However, as expected, because different datasets in the archive contain data with completely different characteristics and physical nature, the optimal value of some hyperparameters (e.g. model dimension) did depend on the dataset.

---

### Official Review · AnonReviewer2 · 2020-10-28
**The paper extends the usage of transformer from univariate to multivariate sequential data**

**Rating:** 4
**Confidence:** 4

**Review:**

The authors targeted an important data format, multivariate time series, and extended the usage of the transformer to this format. The effort is appreciated as multivariate time series data is an important problem while the researches there is limited comparing to other sequences problems, e.g. language. Due to the simple idea of extension and similar works before [1], I expected a higher quality of experiments and presentation of this paper. However, experiments and writing need significant improvement for the next submission.

Major Concerns:

1. The experiment's presentation is confusing and I just picked a few examples from Table 1:
a. There is no standard deviation of the result.
b. Some methods are consistently worse than others, like 1-NN-DTWD comparing to 5-NN-DTWD. I am not sure why they are needed in the table.
c. Averaging the RMSE does not deliver much information, especially when there is no normalization on each dataset. For example, the BeijingPM10 will dominate the averaged result.

2. The structure of writing only has a lot of problems. For example, usually, the fully supervised should be presented before semi-supervised while the paper did in a reverse way. Another problem is the classification result, which seems an important part but the table is shown in the Appendix.

3. The selection of datasets needs more justifications.

Reference:
1. https://arxiv.org/pdf/2001.08317.pdf

---

> ### Author Response · Authors · 2020-11-14
> **Response to and adoption of reviewer's suggestions**
>
> We thank the reviewer for the formatting suggestions, all of which we will adopt in the revised manuscript. We will also add a section in the Appendix with a thorough explanation on how the datasets were selected. Finally, we will show standard deviations for our models. Thus, all voiced concerns by the reviewer will be addressed.
>
> Regarding presentation of results in Table 1:
> with the exception of our models, which achieve consistently top performance across datasets, the performance of all other baseline methods significantly fluctuates across datasets, which necessitates displaying all models. In the reviewer’s example comparison pair, 5-NN-DTWD indeed always outperforms 1-NN-DTWD, however the weaker model in the pair, 1-NN-DTWD, outperforms model 5-NN-ED on some datasets, which in turn outperforms 5-NN-DTWD on some other datasets. To conclude, we don’t find a model in the table that is consistently worse than all other models and should thus be excluded. Additionally, the performance of the NN-DTWD method is typically sensitive on the number of Nearest Neighbors, and including models with 2 parameter values helps to better outline the performance of this model.
>
> Regarding reporting the average Root MSE in Table 1:
> We attempt to illustrate performance differences from different perspectives. We agree that one should not primarily rely on the average RMSE to establish which method is best. This is why in the discussion we rely on the average rank (Table 1) and individual ranks (Table 7), and also we compare our method with the overall second best method: “On average, our models attain approx. 16% lower RMSE than the overall second best model (XGBoost), with absolute improvements varying among datasets from approx. 4% to 36%”. We also show the relative performance difference between our method and whichever method ranks 2nd for each dataset in Table 7. As the reviewer correctly points out, the average RMSE will be primarily affected by the “most challenging” datasets. Despite this limitation, we reported it in Table 1 because we found that in the case of some datasets, many models had very similar RMSE values, while their discrete rank would inevitably strongly differentiate between them (e.g. the RMSE of models with rank 3 and rank 6 can be almost identical). Interestingly, sorting models by average RMSE leads to almost the same list as sorting by average rank, especially with respect to top performers, which indicates that this metric conveys information. However, we will now report a new metric $r_j$ for each model $j$ instead, the "average relative difference from mean" over $N$ datasets (details to follow).
>
> “Due to the simple idea of extension and similar works before[1], I expected a higher quality of experiments”
> The work cited by the reviewer (Wu et al.), which we also cite in the related work section, employs a _full transformer encoder-decoder_ architecture (as the original by Vaswani et al) for the sole purpose of *forecasting* in *univariate* time series, and evaluates it on a single dataset, against 3 reasonably competitive baselines. Instead, we employ a _transformer encoder_ architecture which supports forecasting, imputation, regression and classification for univariate and multivariate time series. Additionally:
>
> - Our proposed method represents an outstanding improvement on the state of the art in the context of time series classification and regression: evaluating it against 11 baselines on 6 different datasets for regression, and against 5 baselines on 11 different datasets for classification, our results clearly show that it performs significantly better than the best currently available methods, while no other method manages to meaningfully differentiate itself from the rest. Despite decades of innovation and meticulously engineered approaches on those problems, it is telling that the second best methods are XGBoost, which does not even take sequence order into account, and ROCKET, which is based on a linear classifier on top of a flat collection of randomly initialized convolutional filters.
> - Our method is the only method which has been shown to successfully leverage unlabeled data in order to improve performance and push the state of the art in time series regression and classification. Leveraging unlabeled time series data is immensely interesting for nearly all domains in the sciences and in industry. Importantly, we show that it can accomplish this even when the number of available unlabeled data points is very limited, and that it can in fact benefit even from reusing labeled data samples through unsupervised learning.
> - These were accomplished by 1) suitably adapting a model architecture which has never before been used for these problems, 2) employing a novel unsupervised objective which allows unsupervised representation learning, 3) developing a framework which allows the same model to be used for different objectives (imputation, forecasting, classification, regression).

---

> > ### Author Response · Authors · 2020-11-14
> > **New metric replacing Average Root MSE**
> >
> > To address the limitations of Average Root MSE (i.e. its bias towards datasets with higher absolute RMSE values),  we will now report a new metric $r_j$ for each model $j$, the "average relative difference from mean" over $N$ datasets: $$ r_j = \dfrac{1}{N}\sum_{i=1}^{N}\dfrac{R(i,j)-\bar{R_i}}{\bar{R_i}}, \quad \bar{R_i} = \dfrac{1}{M}\sum_{k=1}^{M} R(i,k)$$, where $R(i,j)$ is the RMSE of model $j$ on dataset $i$ and $M$ is the number of models.

---

### Author Response · Authors · 2020-11-25
**Additions and updates to revised manuscript**

We have taken into account the feedback that we have received to significantly improve our manuscript.

Specifically, the revised manuscript version includes the following additions, updates and changes:

- We have added a Table in the Appendix comparing performance between different masking schemes used as an unsupervised objective

- We have added a Table in the Appendix comparing the trade-off in performance and training time between allowing all layers of our models to be trainable during fine-tuning, versus freezing all layers except for the output layer.

- We have added a Table in the Appendix comparing performance between using Layer Normalization and Batch Normalization.

- We have added Tables in the Appendix with the variance exhibited in the performance of our models on all datasets.

- We have added Tables in the Appendix with the best hyperparameters we have found for our models on all datasets, including common hyperparameters and a set which performs consistently well across all datasets.

- We have updated the training time comparison table such that we present the *total training times* in seconds (on a GPU for our model). We thus show that our models are computationally tractable.

- We have replaced the Average Root Mean Square Error Metric with the Average Relative Difference from Mean

- We have added criteria/explanations for dataset selection in the Appendix

- We have added a figure in the Appendix showing an example of the performance for imputation on one of the datasets we have considered

- We swapped the order of supervised/unsupervised learning in the Methods section and brought back the Table comparing classification performance from the Appendix to the main body of the paper.

- We have added a reference to a paper using transformers for univiariate, multi-horizon forecasting

- We have clarified the meaning of "regression" in the context of our work

- Several other improvements in phrasing, typos etc

---

### Decision · Program_Chairs · 2021-01-07
**Final Decision**

**Decision:**

Reject

**Comment:**

The authors extends the transformer to multivariate time series. The proposed extension is simple, and lacks novelty. Some design decisions of the proposed method should be better justified. Similar works that also use the transformer for timeseries are not compared.

Experimental results are not convincing. The settings are unclear, and the selection of datasets needs more justifications. Some important experiments are missing.

Finally, writing can also be improved.

---

> ### Comment · ~George_Zerveas1 · 2024-03-14
> **This evaluation has aged well**
>
> This paper was published in KDD in 2021, and as of March 2024 has 644 citations, with significant impact across many domains.
> Not too shabby for an unconvincing paper proposing "a simple extension that lacks novelty".